# GLaVE-Cap: Global-Local Aligned Video Captioning with Vision Expert Integration

## Abstract

Video detailed captioning aims to generate comprehensive video descriptions to facilitate video understanding. Recently, most efforts in the video detailed captioning community have been made towards a local-to-global paradigm, which first generates local captions from video clips and then summarizes them into a global caption. However, we find this paradigm leads to less detailed and contextual-inconsistent captions, which can be attributed to (1) no mechanism to ensure fine-grained captions, and (2) weak interaction between local and global caption generation. To remedy the above two issues, we propose **GLaVE-Cap**, a **G**lobal-**L**ocal **a**ligned framework with **V**ision **E**xpert integration for **Cap**tioning, which consists of two core modules: TrackFusion enables comprehensive local caption generation, by leveraging vision experts to acquire cross-frame visual prompts, coupled with a dual-stream structure; while CaptionBridge establishes a local-global interaction, by using global context to guide local captioning, and adaptively summarizing local captions into a coherent global caption. Besides, we construct **GLaVE-Bench**, a comprehensive video captioning benchmark featuring $5\times$ more queries per video than existing benchmarks, covering diverse visual dimensions to facilitate reliable evaluation. We further provide a training dataset **GLaVE-1.2M** containing 16K high-quality fine-grained video captions and 1.2M related question-answer pairs. Extensive experiments on four benchmarks show that our GLaVE-Cap achieves state-of-the-art performance. Besides, the ablation studies and student model analyses further validate the effectiveness of the proposed modules and the contribution of GLaVE-1.2M to the video understanding community. The source code, model weights, benchmark, and dataset will be open-sourced.

## 1 Introduction

The evolution of vision-language models (VLMs) has marked a significant milestone in multi-modal learning, enabling the joint understanding of visual and textual data (Radford et al., 2021; Li et al., 2023b; Hurst et al., 2024). While early VLMs primarily focused on static image-text tasks, recent efforts have extended this paradigm to the video domain, giving rise to video-language models (Video-LLMs) (Georgiev et al., 2024; Bai et al., 2025). These models are designed to process and reason over temporally extended visual content in conjunction with language, introducing new challenges and opportunities in video understanding. Drawing on insights from the image domain, where fine-grained captions have proven crucial for enhancing visual understanding (Yan et al., 2024; Onoe et al., 2024), recent research has increasingly focused on constructing fine-grained video caption datasets (Chai et al., 2025; Yang et al., 2024; Chen et al., 2024a; Zhang et al., 2024). These fine-grained video captions, which comprehensively describe attributes and motions of all objects and maintain consistency across descriptions, serve as a cornerstone for fine-grained video understanding (Chen et al., 2024a; Zhang et al., 2024).

However, generating detailed video captions remains a significant challenge. Directly applying VLMs to video inputs often leads to the omission of critical visual details (Chen et al., 2024a). A common strategy adopted in recent works is the **local-to-global captioning paradigm**, which first generates local captions for short clips and subsequently summarizes them into a holistic video caption (Yang et al., 2024; Zhang et al., 2024; Chen et al., 2024a). This paradigm decomposes the captioning task, allowing VLMs to process only a small set of visual inputs at one time, helping preserve fine-grained details and improving scalability to longer videos. However, when the local-to-global scheme is

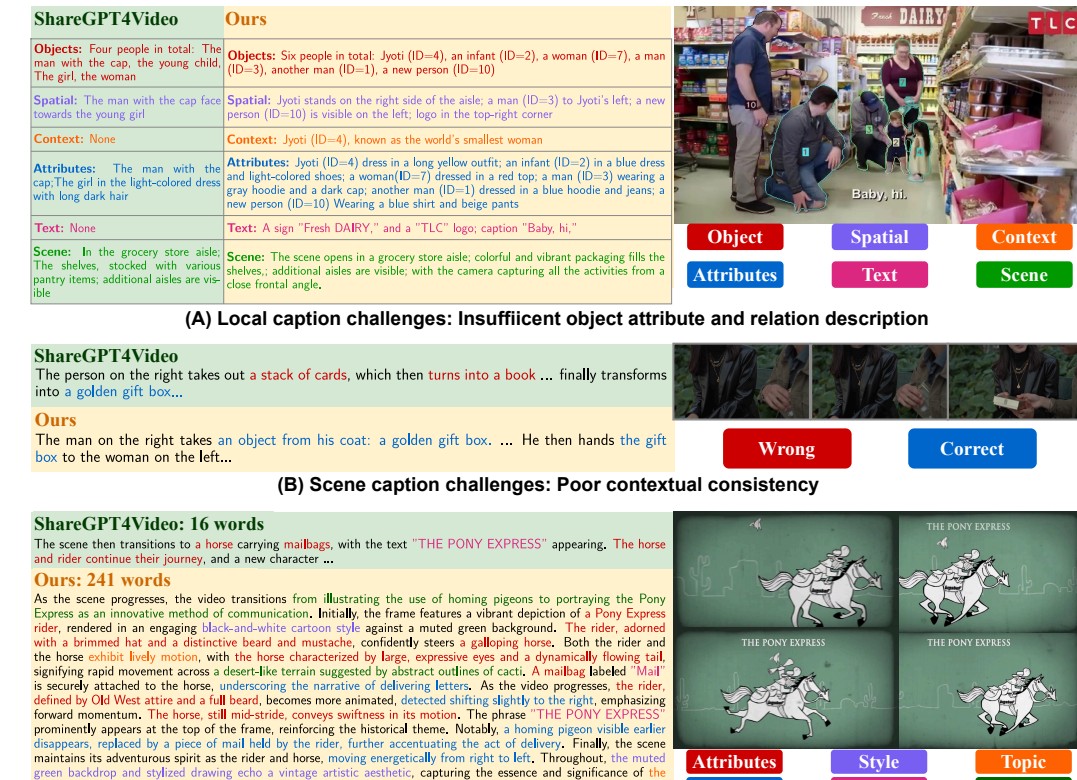

Figure 1: Illustration of the limitations of current methods. Refer to Fig. 15 for full caption of (A).

applied to complex videos involving multiple objects and scenes, these methods commonly face three limitations, as illustrated in Fig. 1: (1) **Insufficient object attribute and relation description**: Local captions often fail to capture challenging attributes such as object spatial relations and quantity, and they tend to be biased toward dynamic changes while overlooking detailed information about object attributes and the scene context. (2) **Poor contextual consistency**: Local captions generated based only on partial video content often struggle to accurately interpret visual elements, and may lead to inconsistencies among multiple local captions, ultimately resulting in a contextually inconsistent video caption. (3) **Information loss during summarization**: Summarizing all local captions into a single video caption in one step is challenging to LLMs, which inevitably omit a substantial amount of detailed information. Some methods attempt to address this issue by multi-step summarization (Yang et al., 2024; Zhang et al., 2024). However, they often lack an effective strategy to integrate these intermediate results into a coherent and comprehensive video caption. Together, the aforementioned issues hinder the generation of accurate and fine-grained video captions.

We attribute the limitations above to two underlying causes. First, local caption generation relies solely on carefully crafted text prompts, which offer limited control over output quality when facing challenging tasks, such as counting. In addition, the sequential nature of keyframe inputs predisposes VLMs to focus on inter-frame dynamics, while static details are frequently overlooked. Second, the limited interaction between local and global caption generation leads to global inconsistencies across segments and substantial semantic loss or redundancy during suboptimal summarization. To tackle these challenges, we propose **GLaVE-Cap**, a **G**lobal-**L**ocal **a**ligned video caption framework with **V**ision **E**xpert integration, which provides highly fine-grained and consistent video caption across both local and global perspectives. Our GLaVE-Cap framework consists of two core modules: TrackFusion and CaptionBridge. **TrackFusion** aims to enhance the comprehensiveness of local captions while ensuring inter-frame consistency. It leverages video-specific expert models (Liu et al., 2024a; Ravi et al., 2024) to track objects across frames, which can better guide VLMs and enrich local annotations. Additionally, to alleviate overemphasis on inter-frame dynamics, it generates local captions via a dual-stream structure, where one stream processes inter-frame dynamics while the other extracts intra-frame static details, thereby effectively preserving the expert-refined information in local

captions. **CaptionBridge** aims to integrate local and global captions. It first generates an overview caption, which can provide contextual guidance to the local captioning process, thereby reducing ambiguity and inconsistency. In parallel, it performs adaptive scene segmentation and summarization based on the semantic content of local captions, thereby ensuring semantic coherence within the scene and preventing information loss and semantic redundancy due to suboptimal summarization.

For evaluating the quality of video captions, a common approach is to assess to what extent a caption can serve as a proxy for the video in answering content-related questions (Chai et al., 2025; Chen et al., 2025). However, existing captioning benchmarks (Chen et al., 2025) suffer from low annotation quality and consist only of single-scene videos, limiting their applicability to evaluate complex video captioning tasks. Moreover, general-purpose video question-answer (VQA) benchmarks (Fu et al., 2025; Li et al., 2024c) typically offer few questions per video, limiting their ability to thoroughly assess caption content. To fill this gap, we construct a manually refined fine-grained Video-QA benchmark **GLaVE-Bench**, which includes multi-scene videos and $5\times$ more queries per video than existing benchmark, covering diverse visual aspects for a comprehensive evaluation of fine-grained video captioning.

We apply GLaVE-Cap on various cutting-edge VLM backbones such as GPT-4o (Hurst et al., 2024) and Qwen2.5-VL-72B (Bai et al., 2025), both achieving state-of-the-art (SOTA) performance on widely used VQA benchmarks such as Video-MME (Fu et al., 2025), MVBench (Li et al., 2024c), and VidCapBench (Chen et al., 2025). These experimental results provide strong evidence for the high quality and generalizability of GLaVE-Cap. In addition, we leverage GLaVE-Cap to generate an extensive training dataset **GLaVE-1.2M**. This dataset includes 16K fine-grained video captions and 1.2M corresponding QA pairs. The GLaVE-1.2M dataset serves as a robust resource for advancing research on fine-grained video understanding capabilities of Video-LLMs. We further train a lightweight student model **GLaVE-7B** on the GLaVE-1.2M dataset and also achieve notable performance, demonstrating the effectiveness of our fine-grained supervision. In a nutshell, we summarize our three main contributions as follows:

- We propose GLaVE-Cap, which can provide highly fine-grained and consistent video captions across both local and global perspectives. It incorporates two key modules: TrackFusion leverages vision experts and a dual-stream design to generate consist and comprehensive local captions; CaptionBridge aligns local captions and performs adaptive scene-level summarization via local-global interaction, ensuring fine-grained and consistent video captions.

- We present GLaVE-Bench, a comprehensive benchmark with $5\times$ more queries per video than existing benchmarks, covering diverse visual dimensions for reliable evaluation of fine-grained video captioning. Additionally, we construct GLaVE-1.2M, a large-scale dataset with fine-grained QA pairs to advance fine-grained video understanding in Video-LLMs.

- We apply GLaVE-Cap to various VLM backbones, achieving SOTA performance on multiple pubic benchmarks and GLaVE-Bench. Additionally, we train a student model, GLaVE-7B, to explore the effectiveness of our fine-grained video captions and QAs.

## 2 RELATED WORK

**Video-Language Models.** Recent years have witnessed remarkable progress in video understanding through the development of large vision language models. Early explorations (Zhang et al., 2023; Lin et al., 2024a; Maaz et al., 2024; Li et al., 2024d; Zhang et al., 2025a) established foundational approaches to align the video and text modalities. Building upon these foundations, recent works (Lin et al., 2024b; Li et al., 2024a; Zhang et al., 2024; Bai et al., 2025; Chen et al., 2024d) further demonstrated that vanilla image language models that support multi-image as input can generalize to video understanding in a zero-shot manner and achieve improved performance when finetuned on video data. Despite variations in design, these models commonly depend on large-scale, high-quality video datasets to improve video understanding.

**Video Captioning Pipeline.** High-quality video text annotations are crucial for modality alignment when training large-scale video-language models. Early approaches (Miech et al., 2019; Zellers et al., 2021; Xue et al., 2022) primarily relied on metadata or ASR-generated subtitles to annotate videos at scale. With the rapid advancement of vision language models (Li et al., 2023b; Zhu et al., 2024; Zhang et al., 2023), Wang et al. (2023); Chen et al. (2024b) explored utilizing these models to generate semantically aligned video captions. However, these works typically focus on short video clips,

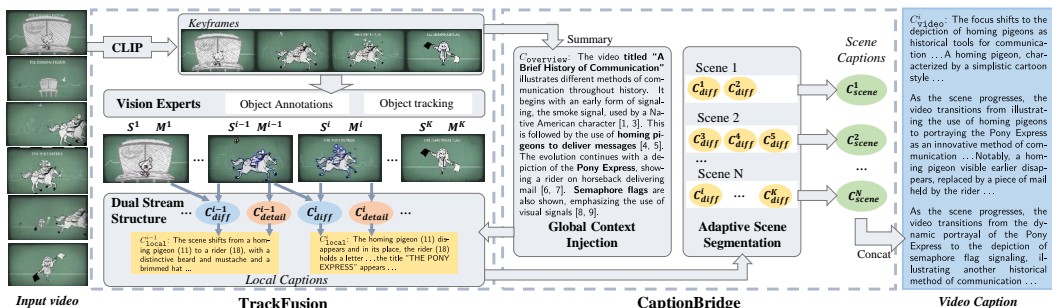

Figure 2: Overview of GLaVE-Cap. Given a video, TrackFusion first integrates vision experts to extract keyframes and track objects, then generates fine-grained local captions in a dual-stream manner. CaptionBridge injects the global context into local captions and achieves adaptive scene segmentation and summarization via their semantic information.

limiting their ability to handle longer and more complex videos and generate dense and fine-grained descriptions. To address this limitation, some recent studies (Yang et al., 2024; Chen et al., 2024a; Zhang et al., 2024; Peng et al., 2024) designed dedicated annotation pipeline for improving both the quality and temporal coverage of captions for longer videos. Although existing methods propose reasonable pipelines and carefully craft text prompts to get accurate video annotations from VLMs, their effectiveness remains limited. VLMs often struggle to capture fine-grained details, maintain contextual coherence, and are prone to hallucinations. In contrast, our approach integrates multiple mechanisms to achieve the best detailed video annotation performance.

## 3 METHOD

In this section, we introduce GLaVE-Cap in three parts, as illustrated in Fig. 2: 1) We first introduce the overview of GLaVE-Cap pipeline in Sec. 3.1; 2) We present TrackFusion in Sec. 3.2, which utilizes vision experts and a dual-stream structure to produce fine-grained local captions; 3) We introduce CaptionBridge in Sec. 3.3, which leverages the local-global interaction mechanism to generate coherent and detailed video descriptions.

### 3.1 PIPELINE OVERVIEW

The overall framework of GLaVE is illustrated in Fig. 2. Given an input video $V$, we first apply CLIP (Radford et al., 2021) to extract keyframes $\mathcal{K}$. The extracted keyframes are then processed by TrackFusion module to obtain object masks by utilizing vision experts. The masked keyframes $\mathcal{M}$ and corresponding textual supplementary information $\mathcal{S}$ are integrated to generate two types of local captions: a differential caption $C_{\text{diff}}$ capturing dynamic changes and a detailed caption $C_{\text{detail}}$ focusing on static attributes. These are then integrated into a unified local caption $C_{\text{local}}$. The CaptionBridge module first generates an overview caption $C_{\text{overview}}$ from the keyframes $\mathcal{K}$ and injects it into the local caption generation process. It then derives scene segmentation $\mathcal{SS}$ based on $C_{\text{local}}$ and $C_{\text{overview}}$ and summarizes each segment with the corresponding scene caption $C_{\text{scene}}$. These scene captions are finally concatenated to form the video caption $C_{\text{video}}$.

### 3.2 TRACKFUSION

VLMs have demonstrated strong capabilities in understanding visual contents and are utilized in current local captioning methods (Chen et al., 2024a; Yang et al., 2024; Zhang et al., 2024) by directly feeding multiple key-frames into the model with only prompt-level in a single-pass. However, VLM still struggle with understanding object spatial relations and quantities (Liu et al., 2023; Li et al., 2023a; Hurst et al., 2024), hindering the completeness and accuracy of the local captions. Furthermore, due to the sequential nature of the keyframe inputs, these single-pass approaches tend to emphasize dynamic visual changes while neglecting fine-grained static information, thereby failing to fully exploit the fine-grained perceptual capabilities. To address these limitations, we introduce TrackFusion, a framework that incorporates vision experts to perform cross-frame consistency

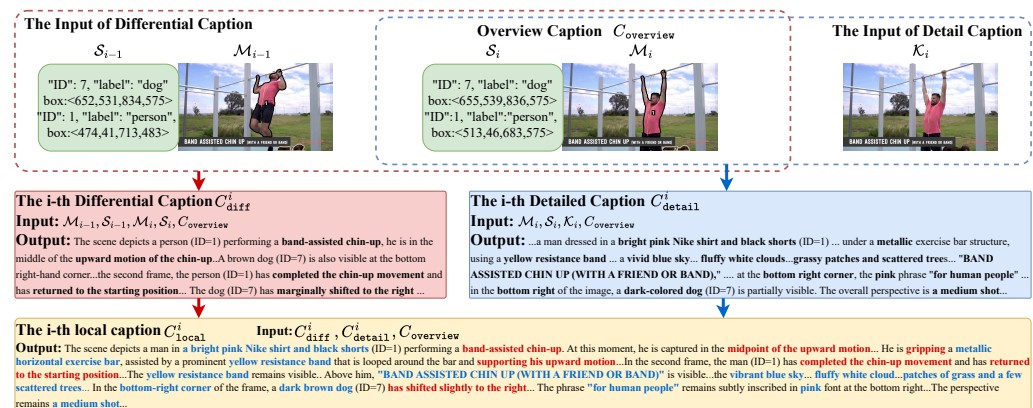

Figure 3: Illustration of the dual-stream structure introduced in TrackFusion.

tracking, enhancing the model's ability to capture object counts and spatial relationships. In addition, we design a dual-stream architecture that separately encodes dynamic variations and static details, and integrates them to generate more comprehensive local captions.

**Track Objects Across Frames Via Vision Experts.** To address the limited capability of capturing and tracking objects in visual contents, recent works introduce image experts to highlight salient objects and generate visual tags (Yang et al., 2023; Yin et al., 2024). However, extending this strategy to video is non-trivial due to frequent object movements, appearances, and disappearances over time. To overcome this challenge, we leverage Grounding DINO (Liu et al., 2024a) and SAM 2 (Ravi et al., 2024) to achieve frame-level object consistency through tracking (We elaborate this design choise in Appendix B). Nevertheless, initializing bounding boxes with Grounding DINO on the first frame and then applying SAM 2's video predictor for subsequent frames struggles to detect and track newly emerging objects. To mitigate this issue, inspired by grounded-SAM 2 (Research, 2024), we integrate Grounding DINO and SAM 2's image predictor to extract bounding boxes $\mathcal{S}$ and object masks for each keyframe. Newly detected objects are assigned a new ID if they have insignificant overlap with existing tracked instances; otherwise, they inherit the ID of the most similar tracked object. The set of image masks with their IDs are then used to update the SAM 2 video predictor, ensuring stable and consistent object segmentation throughout the video. After extracting object masks, for each keyframe, we draw high-contrast boundaries and place numeric IDs at object centers as visual prompts following Set-of-Mark (Yang et al., 2023), generating masked keyframes $\mathcal{M}$ for subsequent local caption generation.

**Dual-Stream Structure.** To overcome the dynamic bias in single-pass local caption generation method, we introduce a dual-stream local captioning structure that decouples the capturing of dynamic and static information, thereby maximizing the detail-capturing capacity of VLMs enhanced by vision experts, as illustrated in Fig. 3. Specifically, we first generate a differential caption $C_{\text{diff}}$ to capture actions and changes across adjacent keyframes. Then, by comparing the original keyframes with their visually prompted counterparts, we produce a detailed caption $C_{\text{detail}}$ focusing on scene context and object attributes. To mitigate object reference ambiguity during merging, the VLM is instructed to append the unique ID of each annotated object when generating both $C_{\text{diff}}$ and $C_{\text{detail}}$. Finally, these two temporary captions are merged into a comprehensive local caption $C_{\text{local}}$. The generation process of local captions can be formulated as follows:

$$C_{\text{diff}}^i = \text{VLM}(\mathcal{M}_{i-1}, \mathcal{M}_i, \mathcal{S}_{i-1}, \mathcal{S}_i, C_{\text{overview}} \mid \text{prompt}_{\text{diff}}) \quad (1)$$

$$C_{\text{detail}}^i = \text{VLM}(\mathcal{K}_i, \mathcal{M}_i, \mathcal{S}_i, C_{\text{overview}} \mid \text{prompt}_{\text{detail}}) \quad (2)$$

$$C_{\text{local}}^i = \text{VLM}(C_{\text{diff}}^i, C_{\text{detail}}^i, C_{\text{overview}} \mid \text{prompt}_{\text{merge}}) \quad (3)$$

where $\mathcal{K}$ and $\mathcal{M}$ denote the original keyframes and their visually prompted counterparts. $\mathcal{S}$ denotes the bounding boxes and $C_{\text{overview}}$ denotes the overview caption, which will be detailed below.

### 3.3 CAPTIONBRIDGE

The existing local-to-global captioning frameworks (Yang et al., 2024; Chen et al., 2024a; Zhang et al., 2024) first generate local captions and then merge the captions into a global description,

ignoring the global context of video, leading to inconsistent descriptions of the same object across video segments. Furthermore, directly summarizing the local captions inevitably omits a substantial amount of detailed information. For the methods involving scene splitting, relying solely on visual changes (Yang et al., 2024; Chen et al., 2024b) may leads to over-segmentation of the video, resulting in incoherent viewpoint changes and redundant descriptions. To address these challenges, we propose CaptionBridge, a bidirectional interaction framework between local and global caption generation. By incorporating a global context into the local caption generation process, we enhance the consistency of local descriptions; additionally, by leveraging the semantic information of local captions, we ensure semantic coherence within the scene, thereby mitigating information loss caused by suboptimal scene segmentation and summarization.

**Global Context Injection.** Current methods typically rely on limited keyframes and the preceding frame-level caption for generating local captions, lacking access to global context and future information. This local-only perspective often leads to inaccurate object recognition and action understanding, resulting in inconsistent video caption, as illustrated in Fig. 1 (B). To alleviate this issue, we introduce an overview caption $C_{\text{overview}}$ that summarizes the global context and inject it into the local captioning process to improve local caption consistency. Specifically, we feed all the keyframes $\mathcal{K}$ into the VLM to generate an overview caption that includes key visual elements, event descriptions, timeline understanding, etc. Additionally, we require the VLM to annotate the corresponding range of keyframes for each sentence in the overview caption, facilitating the localization and understanding during the generation of different captions and summarization:

$$C_{\text{overview}} = \text{VLM}(\mathcal{K}_{[1:n]} \mid \text{prompt}_{\text{overview}}) \tag{4}$$

where $n$ denotes the number of keyframes and $\mathcal{K}_{[1:n]}$ denote all the original keyframes.

**Adaptive Scene-level Segmentation and Summarization.** To address the limitation of inaccurate scene split and summarization mentioned above, we propose an adaptive scene-level summarization strategy that leverages the semantics of local captions to guide the video caption generation process.

Specifically, we first employ PySceneDetect (Breakthrough, 2024) to initialize scene segmentation, which typically produces accurate boundaries but may result in redundant segments due to viewpoint shifts. We then prompt VLM to merge redundant scenes by leveraging the semantic information from all local captions $C_{\text{local}}^{[1:n]}$ and the global context provided by the overview caption $C_{\text{overview}}$. This produces semantically coherent scene partitions with precise boundaries, which serve as the foundation for generating scene-level captions $C_{\text{scene}}$. Finally, by directly concatenating the relatively self-contained scene-level captions in temporal order, we can naturally obtain a comprehensive and fine-grained video-level caption. The overall generation process can be formulated as follows:

$$\mathcal{SS} = \text{VLM}(C_{\text{local}}^{[1:n]}, \text{PSD} \mid \text{prompt}_{\text{SS}}) \tag{5}$$

$$C_{\text{scene}}^i = \text{VLM}(C_{\text{local}}^{[\mathcal{SS}_i.\text{ST}:\mathcal{SS}_i.\text{ED}]}, C_{\text{scene}}^{i-1}, C_{\text{overview}} \mid \text{prompt}_{\text{SC}}) \tag{6}$$

where $\text{PSD}$ is the scene split result of PySceneDetect, $n$ denotes the number of local captions, $\mathcal{SS}_i.\text{ST}$ and $\mathcal{SS}_i.\text{ED}$ denote the start and end keyframe numbers of scene $i$.

# 4 BENCHMARK & DATASET

## 4.1 GLAVE-BENCH

Evaluating the task of video detailed captioning requires a large number of high-quality videos, particularly those featuring complex and multi-scene, along with extensive QA annotations within one video. However, existing datasets either contain only a limited number of scenes (Chen et al., 2024c; Liu et al., 2024b; Li et al., 2024c; Xiao et al., 2021; Chen et al., 2025), making it difficult to assess the effectiveness of video captioning methods in complex scenarios, or include too few questions per video (Chen et al., 2024c; Li et al., 2024c; Xiao et al., 2021; Fu et al., 2025), limiting the comprehensiveness of the evaluation. To fill this gap, we propose GLaVE-Bench, which includes high-quality multi-scene videos and provides over $5\times$ QA pairs per video (118.02 vs. 16.55) than existing benchmark to provide a comprehensive evaluation of video captioning according to Tab. 1. Furthermore, we provide an additional scene hint during evaluation to reduce ambiguity caused by changes in object attributes throughout the video, which is common in multi-scene videos.

Table 1: The comparison of various benchmarks. The number of scenes is estimated by PySceneDetect. Scene-Hints indicates whether scene hints are provided to prevent ambiguity.

| Benchmarks | Videos | Video Length(s) | Scenes/Video | QAs | QA/Video | Annotation | Scene-Hints |
|---|---|---|---|---|---|---|---|
| NExT-QA (Xiao et al., 2021) | 1,000 | 39.5 | 2.95 | 8,564 | 8.56 | Auto | ✗ |
| MVBench (Li et al., 2024c) | 3,851 | 16.0 | 1.43 | 4,000 | 1.04 | Auto | ✗ |
| AutoEval-Video (Chen et al., 2024c) | 327 | 14.6 | 3.37 | 327 | 1.00 | Manual | ✗ |
| TempCompass (Liu et al., 2024b) | 410 | 11.4 | 1.16 | 7,540 | 15.87 | Auto&Manual | ✗ |
| Video-MME-S (Fu et al., 2025) | 300 | 80.7 | 17.67 | 900 | 3.00 | Manual | ✗ |
| VidCapBench (Chen et al., 2025) | 643 | 10.2 | 2.38 | 10,644 | 16.55 | Auto&Manual | ✗ |
| GLaVE-Bench (Li et al., 2024c) | 55 | 85.5 | 8.84 | 6,491 | **118.02** | Auto&Manual | ✓ |

**Scene Hints.** In complex multi-scene videos, a single question may correspond to different answers depending on the specific scene being referenced. Without sufficient scene disambiguation before asking the question, answers can become ambiguous, leading to unreliable evaluation results. To address this, we provide manually verified scene-hints for each scene, which uniquely identify the scene without revealing the answer to any questions. An example illustrating how the absence of scene-hint leads to unreliable evaluation results is presented in Appendix C.2.

**Data Sources and Automatic Annotation.** We manually select 55 videos with multiple scenes, stable camera motion, and well-defined topics from the Video-MME short subset to construct the evaluation set. Question–option pairs are first generated automatically based on fine-grained video captions and scene-level summaries (see Appendix C.1). Each automatically generated QA then undergoes a thorough manual verification process, during which questions are refined or removed to ensure correctness, quality, and appropriate difficulty (see Appendix C.3). This results in a total of 6,491 high-quality QAs. More details and discussions are specified in Appendices C.4 and C.5.

**Evaluation Protocol.** In evaluation, the generated caption serves as a proxy for the video, and is used by an LLM judge (GPT-4o) to answer the benchmark questions. Scene hints from adjacent scenes are additionally provided to reduce ambiguity. If it is insufficient to answer the question with the caption, option "E. Not mentioned" is required to select. Denoting $n_c$, $n_w$ and $n_e$ as numbers of correct, error and "E.", $Acc. = \frac{n_c}{n_c+n_w+n_e}$, $Hall. = \frac{n_w}{n_c+n_w}$, and $N.M. = \frac{n_e}{n_c+n_w+n_e}$ are reported to evaluate accuracy, hallucination ratio, and comprehensiveness of caption, respectively.

## 4.2 GLaVE-1.2M

With the fine-grained video captioning capability of GLaVE-Cap, we further construct a large-scale training dataset GLaVE-1.2M. The video source is LLaVA-Video-178K (Zhang et al., 2024), which comprises 178K high-resolution, untrimmed videos from ten diverse sources. We filter videos based on duration (30–180 seconds) first and then exclude those with less than 2 or over 10 scenes using PySceneDetect. Finally, we assess the videos with GPT-4o (Hurst et al., 2024) to ensure stable camera motion, clear thematic focus, and no frequent scene transitions.

After filtering, we obtain our dataset consisting of 15,814 videos, totaling 376 hours. Using the GLaVE-Cap and QA generation method described in Appendix C.1, we generate a total of 15,814 video captions and 1,176,410 QA pairs. Data distribution of GLaVE-1.2M is presented in Appendix D.

## 5 EXPERIMENTS

### 5.1 VIDEO-CAPTIONING STRATEGY COMPARISON

**Compared methods.** We select LVD-2M (Xiong et al., 2024), AuroraCap (Chai et al., 2025), ShareGPTVideo (Chen et al., 2024a), LLaVA-Video (Zhang et al., 2024), and Vript (Yang et al., 2024) for comprehensive comparison. Among them, ShareGPTVideo and LLaVA-Video adopt innovative approaches: DiffSW and multi-level strategies, which are effective in capturing both fine-grained details and global context. Vript employs a scene-wise summarization strategy, demonstrating stronger temporal consistency. Implementation details of baseline methods are specified in Appendix E.

**Benchmarks & Evaluation.** We evaluate video-captioning strategies on multiple benchmarks, including GLaVE-Bench, VidCapBench (Chen et al., 2025), Video-MME-S (Fu et al., 2025), and five subsets of MVBench (Li et al., 2024c). GLaVE-Bench and VidCapBench primarily focus

Table 2: Quantitative comparison of video captioning strategies. The best results are **bold** and the second-best results are underlined.

| Method | V-L backbone | GLaVE-Bench | | | VidCapBench | | | MME-S | AS | OE | AC | CO | MD |
|---|---|---|---|---|---|---|---|---|---|---|---|---|---|
| | | *Acc.* | *Hall.↓* | *N.M.↓* | *Acc.* | *Pre.* | *Cov.* | *Acc.* | *Acc.* | *Acc.* | *Acc.* | *Acc.* | *Acc.* |
| LVD-2M (Xiong et al., 2024) | Qwen2.5-VL | 42.21 | 19.69 | 47.45 | 13.50 | 51.07 | 83.86 | 64.00 | 53.67 | 48.50 | 38.16 | 73.50 | 31.33 |
| | GPT4o | 37.65 | 19.63 | 53.16 | 12.41 | 49.22 | 81.57 | 59.67 | 55.50 | 47.50 | 37.83 | 69.33 | 25.17 |
| AuroraCap (Chai et al., 2025) | Qwen2.5-VL | 40.00 | 20.93 | 49.41 | 15.94 | 56.08 | 86.60 | 58.93 | 62.83 | 66.16 | 38.33 | 70.83 | 43.67 |
| | GPT4o | 41.16 | 21.71 | 47.42 | 16.01 | 55.84 | 86.34 | 59.79 | 67.50 | 59.17 | 37.67 | 71.83 | 34.67 |
| ShareGPT4Video (Chen et al., 2024a) | Qwen2.5-VL | 47.97 | 19.63 | 40.32 | 16.43 | 56.26 | 88.44 | 65.00 | 57.12 | 65.67 | 39.16 | 73.67 | 42.67 |
| | GPT4o | 45.64 | 17.85 | 44.45 | 16.71 | 56.70 | 87.09 | 63.19 | 62.83 | 63.50 | 38.17 | 79.67 | 44.83 |
| LLaVA-Video (Zhang et al., 2024) | Qwen2.5-VL | 46.56 | 19.65 | 42.06 | 15.47 | 54.49 | 83.67 | 70.33 | 64.83 | 66.67 | 40.33 | 76.67 | 47.17 |
| | GPT4o | 47.88 | 19.02 | 40.88 | 18.25 | 57.55 | 89.07 | 69.37 | 69.00 | 65.00 | 40.83 | 80.17 | 39.33 |
| Vript (Yang et al., 2024) | Qwen2.5-VL | 50.94 | 19.02 | 37.10 | 14.91 | 56.53 | 85.80 | 69.59 | 56.83 | 59.00 | 34.33 | 67.83 | 45.00 |
| | GPT4o | 55.31 | 18.58 | 32.06 | 17.62 | 57.92 | 87.93 | 73.48 | 58.83 | 55.50 | 35.67 | 75.67 | 38.50 |
| GLaVE-Cap | Qwen2.5-VL | 63.67 | 17.42 | 22.90 | 18.78 | **60.98** | **92.34** | 73.19 | 65.50 | **67.17** | 40.33 | 75.83 | 55.50 |
| | GPT4o | **67.68** | **16.04** | **19.40** | **19.40** | 58.61 | 91.79 | **74.52** | **70.67** | 64.33 | **42.17** | **81.17** | **59.33** |

on the granularity of caption descriptions, with GLaVE-Bench being more challenging due to the multi-scene nature of the videos. Video-MME emphasizes the holistic comprehension of multi-scene video content, while MVBench provides a comprehensive evaluation using short video clips.

Following VidCapBench, we evaluate video captions by replacing the original videos with the generated captions and prompting GPT-4o to answer the given question. For GLaVE-Bench, we report *Acc.*, *Hall.*, and *N.M.* following the evaluation protocol described in Sec. 4.1. For VidCapBench, we report *Acc.*, *Pre.*, and *Cov.*. For the remaining VQA benchmarks, we report *Acc.*. We conducted three evaluations for each question to reduce the variability in GPT-based assessments.

Additionally, video re-captioning is widely utilized in video generation, where fine-grained video descriptions are required for high quality video generation. The hallucination accumulation is also a critical concern in the multi-stage video captioning pipeline. Therefore, we evaluate the effectiveness of GLaVE-Cap in video generation setting and perform the hallucination accumulation analysis of GLaVE-Cap for a more comprehensive evaluation. Details are presented in Appendices F and G.

**Experimental Results.** Tab. 2 presents a comparative evaluation of various video captioning strategies. LVD-2M, AuroraCap, and ShareGPT4Video lack well-designed summarization mechanisms, resulting in subpar performance across all four benchmarks. Though Vript performs comparably to our approach on Video-MME owing to its scene-wise summarization strategy, its intra-scene summarization is not comprehensive, resulting in inferior performance on other benchmarks. LLaVA-Video, with its multi-level caption strategy, demonstrates strong capability in VidCapBench and MVBench. However, this approach tends to excessively compress intra-scene details in multi-scene videos, leading to suboptimal performance on GLaVE-Bench and Video-MME. In contrast to the aforementioned methods, GLaVE-Cap achieves SOTA results across all four benchmarks. These results indicate that GLaVE-Cap can effectively handle both fine-grained description (VidCapBench) and visual change capturing (MVBench), also reflecting that the interaction mechanism of CaptionBridge enhances the capability of video understanding and reasoning for multi-scene videos (Video-MME).

Furthermore, we achieve comparable performance when replacing the VLM with the open-source Qwen2.5-VL-72B (Bai et al., 2025). This demonstrates the generalizability of our video-captioning strategy. More aspects of evaluation analysis are specified in Appendix H.

## 5.2 STUDENT MODEL

To evaluate the fine-grained characteristics of our GLaVE-1.2M dataset and its effectiveness in enhancing a model's fine-grained understanding capacity, we train a lightweight model, GLaVE-7B, using Qwen2.5-VL-7B (Bai et al., 2025) as the base model. We then compare GLaVE-7B with the SOTA models (Chen et al., 2024a; Li et al., 2024a; Zhang et al., 2024), on GLaVE-Bench and other widely used VQA benchmarks (Fu et al., 2025; Li et al., 2024c). For a fair comparison, we also train three models based on the Vript (Yang et al., 2024), ShareGPTVideo (Zhang et al., 2025b), and LLaVA-Video (Zhang et al., 2024) datasets using the same base model and training setup, named Qwen2.5-VL-7B-Vript/ShareGPTVideo/LLaVA-Video, respectively. The composition of other datasets used to train Qwen2.5-VL-7B are specified in Appendix I.

Tab. 3 shows that GLaVE-7B achieves the highest performance on GLaVE-Bench, surpassing both prior SOTA models and Qwen2.5-VL-7B fine-tuned on other datasets. On Video-MME and

Table 3: Quantitative comparison of video language models. * denotes our reproduction result.

| Models | GLaVE-Bench | Video-MME-S (Fu et al., 2025) | | MV-Bench (Li et al., 2024c) |
|---|---|---|---|---|
| | *Acc.* | w/o-sub *Acc.* | w-sub *Acc.* | *Acc.* |
| ShareGPT4Video-8B (Chen et al., 2024a) | 51.52 | 39.9 | 43.6 | 51.2 |
| LLaVA-OneVision-7B (Li et al., 2024a) | 72.98 | 58.2 | 61.5 | 56.7 |
| LLaVA-Video-7B (Zhang et al., 2024) | 76.27 | 63.3 | 69.7 | 58.6 |
| Qwen2.5-VL-7B* | 76.31 | 64.26 | 69.52 | 66.80 |
| Qwen2.5-VL-7B-Vript | 75.50 | 61.04 | 65.56 | 66.33 |
| Qwen2.5-VL-7B-ShareGPTVideo | 76.35 | 62.67 | 66.74 | 65.85 |
| Qwen2.5-VL-7B-LLaVA-Video | 75.92 | 60.19 | 64.48 | 67.47 |
| GLaVE-7B | 79.28 | 62.96 | 67.78 | 66.33 |

Table 4: Ablation studies on GLaVE-Bench. The VLM is GPT-4o

| Visual Prompt | Overview Caption | Dual-stream Structure | Adaptive Scene-split | Acc. | Hall.↓ | N.M.↓ |
|---|---|---|---|---|---|---|
| ✓ | ✓ | ✓ | ✗ | 45.98 | 18.53 | 43.38 |
| ✓ | ✓ | ✗ | ✓ | 62.67 | 17.31 | 24.21 |
| ✓ | ✗ | ✓ | ✓ | 66.77 | 16.63 | 19.91 |
| ✗ | ✓ | ✓ | ✓ | 67.09 | 15.90 | 20.22 |
| ✓ | ✓ | ✓ | ✓ | 67.68 | 16.04 | 19.40 |

Table 5: Ablation of visual prompt. VLM: GPT-4o. Metric: *Acc.*

| Dataset / Visual Prompt | ✗ | ✓ |
|---|---|---|
| GLaVE-Bench (Spatial) | 48.13 | 53.14 |
| GLaVE-Bench (Count) | 57.19 | 58.63 |
| GLaVE-Bench (Direction) | 41.89 | 45.42 |
| MVBench (Number count) | 44.17 | 51.33 |
| NExT-QA (Moving count) | 58.07 | 62.94 |

MVBench, GLaVE-7B exhibits a slight performance decline compared to the base model, which we attribute to catastrophic forgetting (Zhai et al., 2023; Li et al., 2024b), a phenomenon also observed in other Qwen2.5-VL-7B fine-tuned models. Notably, the performance decline of GLaVE-7B is consistently smaller than that of other models, indicating that GLaVE-1.2M provides more detailed and comprehensive supervision than other open-source datasets.

## 5.3 ABLATION STUDIES

Tabs. 4 and 5 present the results of our ablation studies. The detailed experimental settings are specified in Appendix J. Removing the dual-stream structure and the adaptive scene split module leads to a notable decrease in accuracy (5.01 and 21.70, respectively), indicating that these components play a crucial role in enhancing the granularity of in-scene descriptions and mitigating information loss during summarization, respectively. Although ablating the overview caption module does not result in a substantial drop in overall performance, it severely compromises the temporal coherence of video captions, thereby implicitly affecting annotation quality, which is not easy to evaluate, as shown in Fig. 1 (b). Although the removal of visual prompts does not significantly impact performance on GLaVE-Bench, Tab. 5 demonstrates that visual prompt ablation causes marked performance degradation in sub-tasks related to object counting across different datasets, suggesting the importance of the integration of vision experts for object quantity perception.

To provide a more comprehensive and accurate understanding of how different components of GLaVE-Cap affect performance, we present more ablations experiments and investigate the influence of prompts and model capabilities. Details are presented in Appendices K to M.

## 6 CONCLUSION

In this paper, we propose GLaVE-Cap, a novel framework for video detailed captioning that addresses the limitations of existing local-to-global paradigms through vision expert integration and local-global interaction. GLaVE-Cap comprises two core modules: TrackFusion, which leverages vision experts and the dual-stream architecture to produce fine-grained, object-aware local captions; and CaptionBridge, which incorporates global context and performs adaptive scene-level summarization to generate coherent and detailed video-level captions. To enable more comprehensive evaluation of video understanding, we introduce GLaVE-Bench, a benchmark that spans diverse visual dimensions. We also release GLaVE-1.2M, a large-scale dataset with high-quality captions and QA pairs to support training and analysis. Extensive experiments demonstrate that GLaVE-Cap achieves state-of-the-art performance across various benchmarks. Further ablation studies and student model analyses validate the effectiveness of each proposed module and highlight the practical utility of GLaVE-1.2M.

**Ethics statement.** 1) Our study engaged a group of volunteer annotators, who contributed to the development of the benchmark and participated in user studies on video quality. These tasks presented no foreseeable risks, and all participants were fully informed of the nature and objectives of their involvement. 2) While the video data in GLaVE-Bench and GLaVE-1.2M is derived from public datasets, we cannot entirely rule out the inadvertent presence of personally identifiable information (PII), such as faces. Users are therefore urged to consult the licensing terms of the original sources and ensure their application of the data adheres to all relevant ethical and legal standards. 3) Similar to other VLMs, our GLaVE-7B may be vulnerable to adversarial prompting that results in harmful or inappropriate outputs. This highlights the ongoing need for advances in AI safety to ensure responsible and secure use of such models.

**Reproducibility Statement.** To ensure the reproducibility of our work, we provide comprehensive resources in the Supplementary Materials. This includes: 1) the complete source code and environment setup scripts for our proposed GLaVE-Cap pipeline; 2) our implementations and setup scripts for baseline video captioning methods; 3) the evaluation code; 4) the full annotations for GLaVE-Bench. Implementation details for the video captioning methods are also elaborated in Appendix E, with detailed training configurations for the student models provided in Appendix I, implementation specifics for the ablation studies in Appendix J, and documentation on the licenses and origins of all used open-source datasets and code in Appendix S.

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

## A  APPENDIX OVERVIEW

In the appendix, we provide more results and showcases. They are structured as follows:

- Justification of using Grounding DINO and SAM-2 in TrackFusion and their advantages over traditional video segmentation models in Appendix B.
- More detailed information and discussions of GLaVE-Bench in Appendix C.
- Visualization of data distributions for GLaVE-1.2M in Appendix D.
- Implementation details of baseline captioning methods in Appendix E.
- Effectiveness of GLaVE-Cap in video generation in Appendix F.
- Hallucination accumulation analysis in Appendix G.
- Additional evaluation analysis in Appendix H.
- Training details of GLaVE-7B in Appendix I.
- Ablation details of GLaVE-Cap in Appendix J.
- More evidence on proposed modules in GLaVE-Cap can address the claimed three issues existing in current methods in Appendix K.
- Analysis of how different types of prompts and capabilities of (M)LLMs affect model performance in Appendix L.
- More ablation studies on other commonly used benchmarks in Appendix M.
- A caption generated by GLaVE-Cap is shown in Appendix N.
- Keyframe Sampled For Fast-changing Vs. Slow Videos are shown in Appendix O
- Prompt templates used in GLaVE-Cap for caption generation in Appendix P.
- Prompt templates used in GLaVE-Bench for evaluation in Appendix Q.
- Discussion of limitations in Appendix R.
- License information in Appendix S.
- Usage of LLM in Appendix T.

## B  SELECTION OF VIDEO SEGMENTATION MODELS IN TRACKFUSION

Most traditional video segmentation models, such as XMem (Cheng & Schwing, 2022) and also SAM-2, do not support textual input. These models typically rely on initial object masks or visual continuity cues and lack semantic understanding. Therefore, they are unsuitable for our task, which requires segmenting and tracking objects based on category prompts that may appear or disappear throughout the video, i.e., Referring Video Object Segmentation (RVOS).

Some RVOS methods (Li et al., 2025; Lai et al., 2024) enhance semantic understanding by fine-tuning the SAM's mask decoder. However, this compromises generalization and tracking stability, and the base model SAM performs notably worse than SAM-2. More recent work, like SAMWISE (Cheng & Schwing, 2022), is the first to integrate textual knowledge into SAM-2. However, **it only supports one object per prompt**, which is incompatible with our need to segment multiple category-level objects simultaneously.

Therefore, we adopt Grounding DINO and SAM-2 for segmentation, as both have been shown to be powerful and reliable in recent works. GLaVE-Cap is also easily extensible to incorporate newer, more advanced models if they are available.

## C  MORE INFORMATION OF GLAVE-BENCH

### C.1  QUESTION-OPTION PAIRS GENERATION

After generating video captions using GLaVE-Cap, we first construct question-answer pairs. Following the 16 question types defined in LLaVA-Video (Zhang et al., 2024), we categorize them into two

Figure 4: Illustration of the example video `6EIrArTyLVU.mp4` in GLaVE-Bench.

groups: scene-level QA and global-level QA. The scene-level QA targets fine-grained understanding within individual scenes, whereas the global-level QA assesses holistic comprehension and reasoning across the entire video.

For scene-level QA, we focus on 12 description-oriented question types, such as description-Object, fine-grained action, and count. For each scene $i$, we generate at most one question–answer pair per type based on the corresponding scene caption $C^i_{\text{scene}}$, ensuring broad coverage of nearly all scenes and their specific content. To reduce ambiguity, we adopt the scene hint strategy introduced in VRIPT-RR (Yang et al., 2024), generating a scene hint for each scene and prepending it to the corresponding question. In addition, we introduce a new question type, Visual-cue, which targets subtle yet informative visual elements—such as text, logos, and other cues—that are crucial for accurately understanding the video content.

For global-level QA, which comprises the remaining four question types targeting temporal order and causal reasoning, we generate up to five question–answer pairs per type based on the full video caption $C_{\text{video}}$. As these question types demand an understanding of long-range dependencies across the video, they can only be effectively constructed from the global context provided $C_{\text{video}}$.

After generating the initial question–answer pairs, we observed that some were not atomic, *i.e.*, they could be decomposed into multiple independent QA pairs. Additionally, since the QA pairs were derived directly from captions, the questions and answers often exhibited strong continuity, with many questions implicitly revealing or suggesting their corresponding answers. To mitigate these issues, we employed GPT-4o (Hurst et al., 2024) to refine the QA pairs by splitting non-atomic questions and rephrasing them to ensure greater neutrality and reduced answer leakage. **The refinement step described in this paragraph was skipped during the construction of GLaVE-1.2M.**

Finally, we generate the question–option pairs. In addition to the refined QA pairs, we provide GPT-4o (Hurst et al., 2024) with the corresponding scene or video caption as contextual guidance. Based on this input, GPT-4o is asked to produce four answer options for each question: one correct answer and three distractors. The distractors are designed to be stylistically similar to the correct answer but factually incorrect, ensuring the plausibility of all options while maintaining a single ground-truth answer.

## C.2 EXAMPLES OF SCENE-HINTS IN GLAVE-BENCH

For instance, to illustrate how the absence of scene-hint can lead to unreliable evaluation results, consider the video `6EIrArTyLVU.mp4` in our benchmark, which is shown in Fig. 4. Without any scene hint, the 55-th question: *"In the early stage of the scene, compared to the man who is exercising, where is the person in the background located?"* could be interpreted differently depending on the referenced scene.

- If **scene 0** is considered, the answer is: *"The person is directly behind the man, using gym equipment in the background."*

- If **scene 5** is considered, the answer becomes: *"The person is located on the left side of the scene, walking across the rubber floor."*

- For the rest of the scenes, the answer should be: *"The man who is exercising is the only person."*

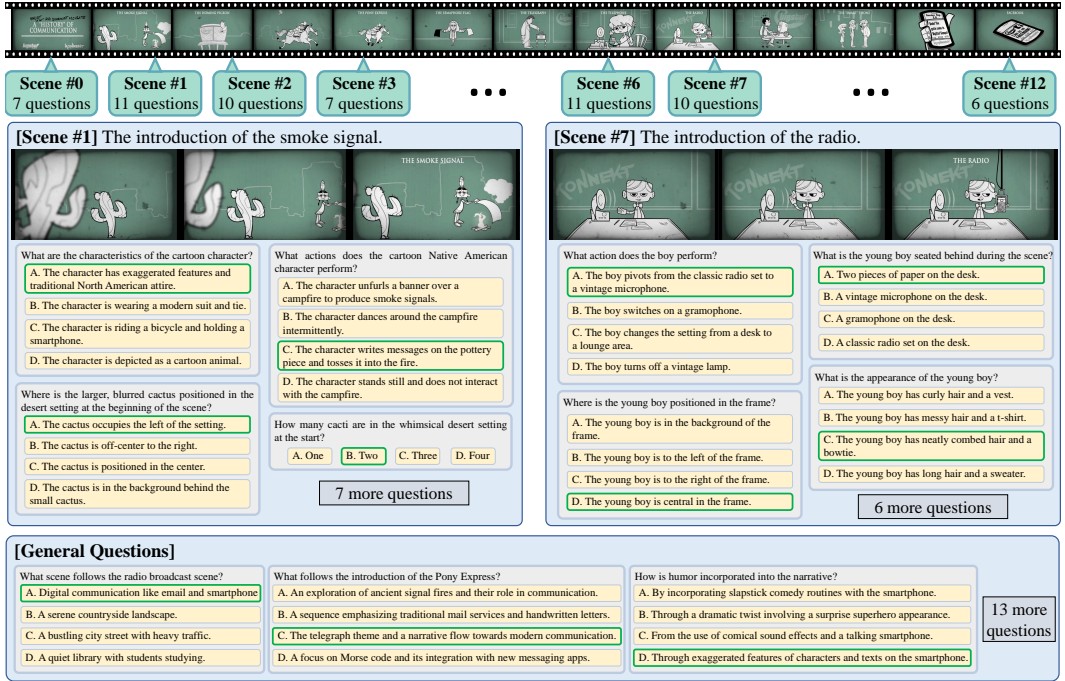

Figure 5: A case of GLaVE-Bench. The video contains 13 scenes and 109 scene questions in total.

To address this, we provide manually verified scene-hints for each scene, which uniquely identify the scene without revealing the answer to any downstream questions. These hints help eliminate ambiguity while preserving fairness and answer neutrality. For example, the scene-hints for this video are:

• 0: A man is introducing something in front of the camera.

• 1: A man is doing pull-ups.

• 2: A man is doing dips on a bar.

• 3: A man is doing tricep dips.

• 4: A man is doing horizontal pulls.

• 5: A man is doing sit-ups.

• 6: A sign is displayed in the center.

These scene-hints ensure reliable evaluation by anchoring the question to the intended context, without leaking potential answers.

## C.3 HUMAN REVIEW

After the initial automatic generation, we incorporate human verification to ensure the accuracy and overall quality of the data. Human annotators are instructed to carry out two main types of validation tasks. First, they verify the scene segmentation and associated hints, checking whether the segmentation is reasonable, whether the scene hints clearly refer to the intended scenes, and whether any unnecessary scene details are revealed. Revisions are made when segmentation or hints are found to be inappropriate. Second, annotators evaluate each question–option pair to verify the correctness of the question–answer pairs, eliminate ambiguity, and ensure consistency between the scene-level QA pairs and the referenced scene. When issues are identified, the QA pairs are either revised or removed. Note that the tasks mentioned above represent typical examples of the verification process; the full annotation guidelines are provided in the supplementary material: Annotation.md. After human verification, a total of 6,491 question–option pairs are retained, with a total of 230 rater-hours spent.

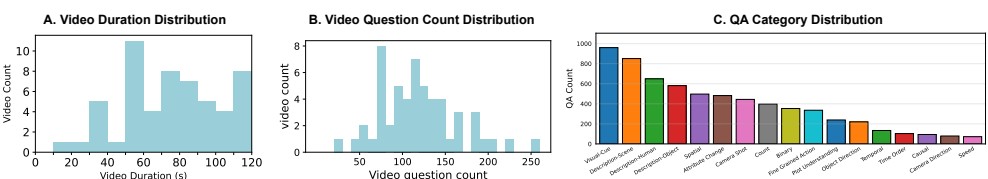

Figure 6: Visualization of various distributions in the test benchmark.

## C.4 Showcase & Visualization of data distributions

Fig. 5 illustrates a showcase of GLaVE-Bench containing 13 scenes and 109 scene questions in total. The sample includes the keyframes, scene hints, questions, answer options, and correct answers for Scenes #1 and #7, alongside three general questions for the whole video.

Fig. 6 (A) shows the distribution of video durations in GLaVE-Bench: longer videos (60–120s) are more prevalent, positioning it as a challenging dataset characterized by longer average video duration like Video-MME Fu et al. (2025). Fig. 6 (B) presents the distribution of video-question counts, which approximately follows a normal distribution. Fig. 6 (C) reports the distribution of 17 question types in GLaVE-Bench. The distribution of question types is largely uniform. The 17 predominant categories each contain 100 to 1k instances, while more challenging ones like Camera Direction and Speed are less frequent. This balance ensures comprehensive coverage during evaluation. Among them, Visual cue, Description-Scene, Description-Human, and Description-Object are the four most frequent types, further emphasizing the benchmark's focus on fine-grained understanding.

## C.5 Fairness, Fidelity and Diversity of GLaVE-Bench

We discuss the fairness, fidelity of details, and video diversity of GLaVE-Bench in this section.

Firstly, although initial question–answer pairs are generated via the GLaVE-Cap pipeline, GLaVE-Bench's fairness is ensured through three mechanisms. First, all QA pairs underwent manual refinement to guarantee correctness and to reduce omissions of salient video content. Second, by leveraging GPT-4o to extract objects and descriptive attributes from each video, we found that the captions used to build GLaVE-Bench contain on average 133 unique mentions of characters, objects, scenes, and their attributes. This richness supports the creation of 118 QA pairs per video that together provide a comprehensive reflection of the original material, thereby mitigating bias or distortion. Third, we do not evaluate models using the exact captions from which QA pairs were derived; instead, captions are regenerated at evaluation time (see Table Tab. 2), ensuring that reported performance is reproducible and not artificially inflated by benchmark construction artifacts.

Secondly, GLaVE-Bench maximizes detail recall through the GLaVE-Cap process and then increases detail precision via manual refinement. The challenge of achieving detailed coverage is a common limitation across existing benchmarks. In our comparison with datasets such as Video-MME and MVBench, we observed that these benchmarks are limited in their capacity to assess fine-grained video understanding—an issue GLaVE-Bench is designed to address. To further reduce omissions arising from caption-based QA generation, we manually verified and refined all question–answer pairs, ensuring that important content omitted during automatic captioning is recovered.

Finally, the videos of GLaVE-Bench span all six major categories defined by Video-MME (i.e., Knowledge, Film & Television, Sports Competition, Artistic Performance, Life Record, and Multi-language), providing broad topical coverage, which ensure the diversity of video source. We also prioritize the density and diversity of QA per video over simply increasing the number of videos, addressing the issue of missing fine-grained details and visual content in previous benchmarks (e.g., Video-MME).

# D Visualization of Data Distributions in GLaVE-1.2M

Fig. 7 (A) illustrates the detailed distribution of six major categories and thirty subcategories in GLaVE-1.2M, where the categories are fully aligned with those of Video-MME. For most common categories, GLaVE-1.2M contains more than 500 video samples. Notably, the Daily life subcategory,

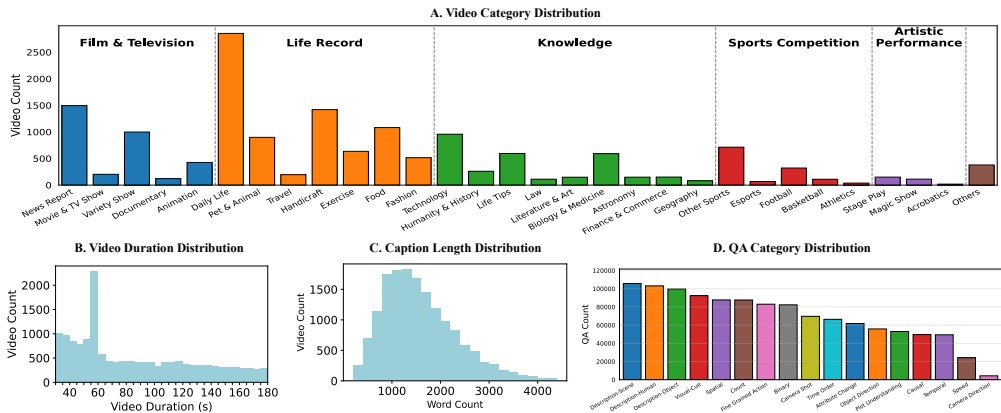

Figure 7: Video category distribution, video duration distribution, caption length distribution and QA category distribution in GLaVE-1.2M.

which covers the broadest range of content, includes nearly 3,000 videos. The relatively rare categories, such as Law and Stage play, also contain around 100 videos each, ensuring diversity within the training data. Fig. 7 (B) shows the distribution of video durations in GLaVE-1.2M: shorter videos (0–60s) are more prevalent, while those between 60–180s exhibit a relatively balanced distribution. Fig. 7 (C) presents the distribution of fine-grained video description lengths, which approximately follows a normal distribution. Fig. 7 (D) reports the distribution of 17 question types in GLaVE-1.2M. The distribution of question types is largely uniform. The 15 predominant categories each contain 50k to 100k instances, while more challenging ones like Camera Direction and Speed are less frequent. This balance ensures comprehensive coverage during training. Among them, Description-Scene, Description-Human, Description-Object, and Visual cue are the four most frequent types, further emphasizing the dataset's focus on fine-grained understanding.

## E  IMPLEMENTATION DETAILS OF BASELINE CAPTIONING METHODS

We selected several representative video captioning pipelines proposed in previous works. As many of these works did not release their complete pipeline code, we implemented the methods based on the descriptions provided in their respective papers. In our reproductions, we preserved the original prompt templates, frame sampling strategies, and multi-step annotation procedures whenever available. However, for a fair comparison, we standardized certain components across all methods, such as the choice of annotation model and the input image resolution.

In the following, we outline the implementation details of each baseline pipeline used in our experiments.

- **LVD-2M** (Xiong et al., 2024): This method generates a detailed caption for every 30-second video segment by sampling 6 frames as input to the annotation model, followed by refinement and summarization. In our implementation, the 6 frames are provided as separate high-resolution images, rather than as a single $2 \times 3$ grid image as originally reported.

- **AuroraCap** (Chai et al., 2025): This method first produces a structured caption and then refines it into a detailed caption. Since the original paper did not specify a frame sampling strategy, we chose to sample at 1 frame per second, or use a fixed 32-frame sampling strategy for videos longer than 32 seconds.

- **ShareGPT4Video** (Chen et al., 2024a): This work proposes the DiffSW captioning pipeline, which we implemented faithfully according to the description provided in the paper.

- **LLaVA-Video** (Zhang et al., 2024): This method introduces a three-stage aggregative captioning pipeline. We reproduced the pipeline using the official code released by the authors.

- **Vript** (Yang et al., 2024): This method divides the video into multiple clips and generates captions for each clip individually. The original implementation includes audio transcripts as input during

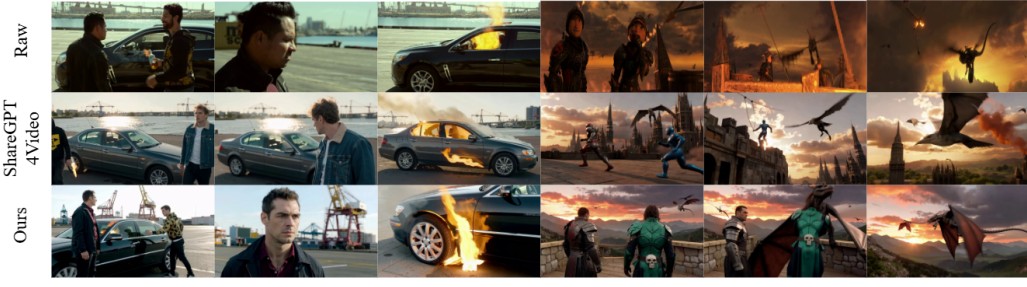

Figure 8: Illustration of four videos generated by captions of GLaVE-Cap and ShareGPT4Video.

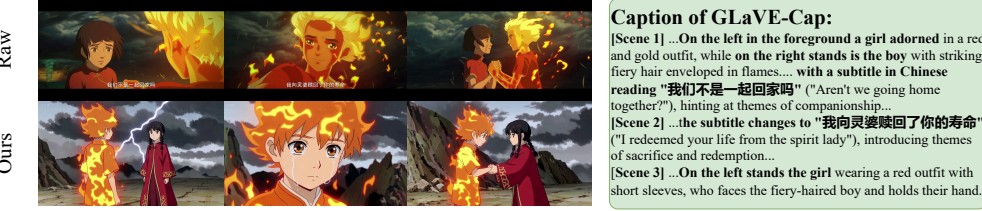

**Caption of GLaVE-Cap:**
[Scene 1] ...**On the left in the foreground a girl adorned** in a red and gold outfit, while **on the right stands is the boy** with striking fiery hair enveloped in flames.... **with a subtitle in Chinese reading "我们不是一起回家吗"** ("Aren't we going home together?"), hinting at themes of companionship...
[Scene 2] ...**the subtitle changes to "我向灵婆赎回了你的寿命"** ("I redeemed your life from the spirit lady"), introducing themes of sacrifice and redemption...
[Scene 3] ...**On the left stands the girl** wearing a red outfit with short sleeves, who faces the fiery-haired boy and holds their hand.

Figure 9: Illustration of an example video and its corresponding caption generated by GLaVE-Cap. The bold font indicates content in the caption which is not successfully generated.

caption generation, but we excluded it in our reproduction as audio is beyond the scope of this work.

## F    EFFECTIVENESS OF GLAVE-CAP IN VIDEO GENERATION

we selected 15 short video clips and generate captions using both our GLaVE-Cap and ShareGPT4Video. These captions were then used as prompts in Seedance (Gao et al., 2025) to generate videos. The selected videos cover diverse content, including 8 human-centric, 3 landscape, 2 object-focused, and 2 animal-related clips, with 11 realistic and 4 animated in style.

As demonstrated by Fig. 8, videos generated using captions from GLaVE-Cap consistently outperform those based on ShareGPT4Video captions across diverse video content. This superiority is manifested in several aspects: more authentic transitions in *video (A)*, *video (C)*, more accurate character attributes in *video (A)* and *video (B)*, and better-aligned plot coherence in *video (D)*. Furthermore, our experiments reveal that GLaVE-Cap successfully captures subtle details from the original videos, such as the on-screen text and character positioning in *video A*. However, these fine-grained details were not faithfully reproduced by the video generation models, as illustrated in Fig. 9.

Furthermore, We conducted a user study to assess the potential of our captions for guiding video generation. Specifically, We ask users to compare which one better preserved the original video's storyline, scene structure, and visual details on a scale of 1-5. 1/5 means ShareGPT4Video/Ours outperforms the other in almost all aspects, 2/4 means ShareGPT4Video/Ours outperforms the other in most aspects, but the other still had a slight advantage, and 3 means a tie. We also randomize the presentation order of the two generated videos for each source video to ensure fairness. We collected

370 valid responses. Among them, videos generated from GLaVE-Cap captions were preferred in 72.16% of the cases, tied in 10.27%, and underperformed in 17.57%. Moreover, in 45.67% of the comparisons, GLaVE-Cap was rated 5/5, indicating that our generated videos outperformed those of ShareGPT4Video across nearly all evaluated aspects.

These observations and user study results indicate that the captions produced by our GLaVE-Cap can provide more detailed and accurate video-text alignment annotations, thereby exhibiting significant potential to enhance the performance of video generation models. We also plan to further explore the use of GLaVE-Cap-captioned datasets to fine-tune video generation models, especially for complex long-form video synthesis, in the further works.

## G  HALLUCINATION ACCUMULATION ANALYSIS

Video detailed captioning typically involves multiple steps (Zhang et al., 2024; Chen et al., 2024a), and our framework follows a similar multi-stage process. However, most steps in our framework are pure-text summarization processes, which significantly limit the propagation of hallucinations. Notably, GPT-4o and Qwen2.5-72B, the backbone we adopted, achieve only 1.5% and 4.3% hallucination rate on the Hughes Hallucination Evaluation Model (HHEM) leaderboard[1], indicating high reliability for text-only tasks. The main source of potential hallucination arises from extracting dynamic changes and static details from keyframes, which inherently involves vision-language interaction and is difficult to avoid in the video detailed captioning task. To reduce hallucination in this stage, we have adopted visual prompts provided by expert models. This strategy has been shown to be effective in minimizing hallucinations in prior works such as Set-of-Mark (Yang et al., 2023).

To further evaluate potential hallucination accumulation, we conduct a quantitative evaluation on the semantic detail subset of *VideoHallucer*(Wang et al., 2024). We follow the caption evaluation protocol introduced in the main paper (same as Table 2). Our method achieves a higher positive accuracy while maintaining hallucination accuracy comparable to baseline methods (see Table 6). It demonstrates that our pipeline does not suffer from noticeable hallucination accumulation.

Table 6: Comparison on the semantic detail subset of *VideoHallucer*.

| Method | Basic ↑ | Hallucinated ↓ | Overall ↑ |
|---|---|---|---|
| LVD-2M (Xiong et al., 2024) | 52.0 | 96.0 | 49.0 |
| AuroraCap (Chai et al., 2025) | 41.0 | 93.0 | 35.5 |
| ShareGPT4Video (Chen et al., 2024a) | 57.0 | 93.0 | 52.0 |
| LLaVA-Video (Zhang et al., 2024) | 59.5 | 94.5 | 55.5 |
| Vript (Yang et al., 2024) | 60.5 | 95.0 | 58.0 |
| **Ours** | 62.0 | 93.5 | 56.5 |

## H  ADDITIONAL EVALUATION ANALYSIS

### H.1  SAMPLE FRAME COUNT COMPARISON

As discussed in Appendix E, we did not unify the video frame sampling strategy between captioning methods, since some methods lack support for flexible frame sampling and the effect of frame count on caption quality remains uncertain. To improve the credibility of our comparison, we report the average number of frames used for caption generation in Tab. 7. Some methods process more input frames, theoretically accessing more video information, but still fail to produce better captions.

### H.2  EVALUATION STABILITY

In Sec. 4.1, we introduce an automatic evaluation strategy for GLaVE-Bench, where GPT-4o serves as the judge model to select the best answer based on the captions under evaluation. However, since GPT-4o may exhibit instability in its judgment, this section focuses on analyzing the stability of the

---

[1] https://huggingface.co/spaces/vectara/leaderboard

Table 7: Average sample frame count comparison across multiple methods and benchmarks

| Methods | GLaVE-Bench | MME | AS | OE | AC | CO | MD | VidCap |
|---|---|---|---|---|---|---|---|---|
| LVD-2M (Xiong et al., 2024) | 15.80 | 16.15 | 6.11 | 1.00 | 4.37 | 5.41 | 1.00 | 2.00 |
| AuroraCap (Chai et al., 2025) | 31.57 | 31.56 | 27.88 | 5.00 | 21.02 | 26.01 | 5.00 | 10.07 |
| ShareGPT4Video (Chen et al., 2024a) | 30.19 | 32.06 | 11.93 | 2.53 | 7.64 | 9.18 | 2.48 | 5.26 |
| LLaVA-Video (Zhang et al., 2024) | 79.34 | 81.08 | 27.88 | 6.00 | 22.34 | 27.56 | 6.00 | 10.46 |
| Vript (Yang et al., 2024) | 42.90 | 59.06 | 6.69 | 3.00 | 4.54 | 4.50 | 3.00 | 8.21 |
| GLaVE-Cap (Ours) | 30.19 | 32.06 | 11.93 | 2.53 | 7.64 | 9.18 | 2.48 | 5.26 |

Table 8: Results on GLaVE-Bench across three runs.

| Method | V-L backbone | Run 1 | | | Run 2 | | | Run 3 | | |
|---|---|---|---|---|---|---|---|---|---|---|
| | | Acc. | Hall.↓ | N.M.↓ | Acc. | Hall.↓ | N.M.↓ | Acc. | Hall.↓ | N.M.↓ |
| LVD-2M (Xiong et al., 2024) | Qwen2.5-VL | 41.73 | 19.54 | 48.13 | 42.23 | 20.23 | 47.07 | 42.66 | 19.29 | 47.14 |
| | GPT4o | 37.36 | 19.60 | 53.54 | 38.08 | 19.92 | 52.44 | 37.50 | 19.38 | 53.49 |
| AuroraCap (Chai et al., 2025) | Qwen2.5-VL | 40.04 | 20.59 | 49.58 | 40.02 | 20.96 | 49.36 | 39.93 | 21.24 | 49.30 |
| | GPT4o | 40.86 | 21.68 | 47.84 | 41.46 | 21.50 | 47.19 | 41.18 | 21.96 | 47.23 |
| ShareGPT4Video (Chen et al., 2024a) | Qwen2.5-VL | 47.93 | 19.74 | 40.29 | 47.90 | 20.06 | 40.09 | 48.08 | 19.08 | 40.58 |
| | GPT4o | 45.39 | 18.23 | 44.49 | 45.49 | 17.88 | 44.60 | 46.03 | 17.44 | 44.25 |
| LLaVA-Video (Zhang et al., 2024) | Qwen2.5-VL | 46.63 | 19.26 | 42.24 | 46.48 | 20.06 | 41.86 | 46.56 | 19.63 | 42.07 |
| | GPT4o | 47.57 | 19.77 | 40.70 | 48.04 | 18.76 | 40.87 | 48.02 | 18.53 | 41.06 |
| Vript (Yang et al., 2024) | Qwen2.5-VL | 50.56 | 19.26 | 37.37 | 51.21 | 18.75 | 36.97 | 51.04 | 19.06 | 36.94 |
| | GPT4o | 55.28 | 18.77 | 31.95 | 55.42 | 18.55 | 31.97 | 55.25 | 18.43 | 32.28 |
| GLaVE-Cap (Ours) | Qwen2.5-VL | 63.93 | 17.03 | 22.94 | 63.26 | 17.90 | 22.95 | 63.83 | 17.32 | 22.80 |
| | GPT4o | 67.68 | 16.00 | 19.43 | 67.69 | 16.02 | 19.40 | 67.66 | 16.09 | 19.37 |

evaluation process. Specifically, we conduct each evaluation three times for every method and assess the consistency of the results. As shown in Tab. 8, the calculated metrics exhibit minimal fluctuation. Furthermore, we categorize each question into three types based on the consistency of the results:

- **Consistent**: All three evaluations yield the same result. This indicates a reliable and stable judgment by GPT-4o. We further divide this type into three sub-categories: "correct", "wrong", and "not mention".

- **Inconsistent due to 'E'**: The three results are not identical, but the variation only occurs between the label "E" (i.e., Not Mentioned) and one specific non-"E" option. This typically happens when the required information is only implicitly stated or must be inferred from the caption. The boundary between selecting "E" or a concrete answer is often ambiguous, leading to this kind of inconsistency. Despite this, such evaluations are still reliable to some extent, and has a reasonable influence to the metrics.

- **Fully Inconsistent**: The results differ across more than two non-"E" options, reflecting a higher degree of instability in the evaluation process.

We visualize the proportion of each category in Fig. 10. It can be shown that most of the evaluations fall into the **Consistent** category, indicating that GPT-4o generally provides stable judgments. Notably, the accuracy improvement achieved by our method is primarily attributed to the "Consistent" category – Our method yields confident and consistent "correct" judgments in 10%-20% questions that are answered 'E' by other methods. This confirms that our performance gain is a reliable reflection of the caption quality.

## I  TRAINING DETAILS

All models, including GLaVE-7B and those fine-tuned on other datasets, are trained based on the Qwen2.5-VL-7B model (Bai et al., 2025). The training data for each model consists of two parts: image data and video data. The image data is identical across all models, comprising 405,752 general-category samples from LLaVA-One-Vision (Li et al., 2024a). The video data, on the other hand, differs depending on the specific dataset used for fine-tuning. The detailed composition of the video data for each model is provided below:

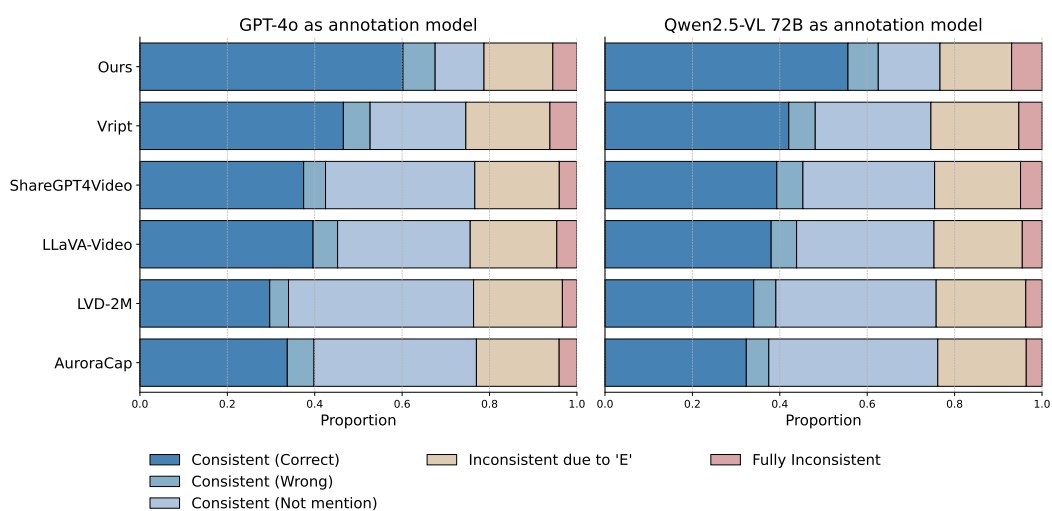

Figure 10: Question consistency type distribution across different methods and base models

- **Vript**: A total of 408,816 video-caption pairs, including 8,776 samples from short videos and 400,040 from long videos.
- **ShareGPT4Video**: Contains 255K samples, including 240,000 open-ended QA items and 15,000 caption entries. The subset selection follows the same criteria as used in the LLaVA-Video model's `llava-hound` training set.
- **LLaVA-Video-178K**: Composed of 5 sources, identical to the training set of the LLaVA-Video model:
    - LLaVA-Video-178: 178,510 caption entries, including 960,792 open-ended QA items and 196,198 multiple-choice QA items
    - NeXT-QA: 17,090 open-ended QA items and 17,024 multiple-choice QA items
    - ActivityNetQA: 23,530 open-ended QA items
    - PerceptionTest: 1,803 open-ended QA items
    - LLaVA-Hound: 240,000 open-ended QA items and 15,000 caption entries.
- **GLaVE-1.2M**: Consists of 1,146,818 open-ended QA items. Caption data is excluded from training due to its small volume and excessive length, which may negatively impact learning.

Training is performed on 64 Ascend 910B2 devices distributed across 8 nodes (8 devices per node, each has 64G memory), using the MindSpeed-MM framework. We follow the official training configuration provided in the MindSpeed-MM repository[2]. The only change is that we set GRAD_ACC_STEP to 16, resulting in a global batch size of 1,024. The training was conducted for a total of 2,000 iterations, taking approximately 60 hours to complete.

## J  ABLATION SETTINGS

The detailed ablation settings are as follows:

- **w/o visual prompt**: We replace the visual-masked inputs $\mathcal{M}_{i-1}$ and $\mathcal{M}_i$ in Eq. 1, 2 with the original keyframes $\mathcal{K}_{i-1}$ and $\mathcal{K}_i$ and remove textual supplementary information $S$ from the input. We also remove all phrases that are related to visual prompts and textual supplementary information in the corresponding prompts. The rest of the architecture remains unchanged.
- **w/o overview caption**: We remove the overview caption input from Eq. 1-3, 6, along with all the overview-related phrases in the corresponding prompts. The rest of the architecture remains unchanged.

---

[2]https://gitee.com/ascend/MindSpeed-MM/blob/master/examples/qwen2.5vl/finetune_qwen2_5_vl_7b.sh

- **w/o dual-stream structure**: We directly use the differential caption $C_{\mathtt{diff}}^i$ as the local caption $C_{\mathtt{local}}^i$, without merging it with the detailed caption $C_{\mathtt{detail}}^i$. The subsequent scene-level summarization strategy remains unchanged.

- **w/o adaptive scene-split**: We treat the entire video as a single scene, *i.e.*, $len(\mathcal{SS}) = 1$ with $\mathcal{SS}_1.\mathtt{ST} = 1$ and $\mathcal{SS}_1.\mathtt{ED} = n$, where $n$ is the number of keyframes. Then, we perform the summary using the standard scene-level summarization approach. The local caption generation strategy remains unchanged.

# K   DIRECT EVALUATION OF MODULE EFFECTS

We conduct further analyses on the generated captions for all 55 videos of *GLaVE-Bench*.

## K.1   TRACKFUSION ADDRESSES INSUFFICIENT OBJECT ATTRIBUTE AND RELATION DESCRIPTION.

To demonstrate this, we compare GLaVE-Cap with an ablated version that removes the dual-stream structure, by leveraging GPT-4o to count the number of direct visual description words (e.g., attributes and interactions of scenes, objects, and characters) in both local and video captions. Tab. 9 shows the average word counts, calculated across the 55 videos.

Table 9: Average number of descriptive words in local and video captions.

| Method | Local | | | | Video | | | |
|---|---|---|---|---|---|---|---|---|
| | #Scene | #Object | #Person | Total | #Scene | #Object | #Person | Total |
| w/o Dual-stream | 4402 | 4924 | 4905 | 14229 | 41 | 38 | 33 | 113 |
| GLaVE-Cap | 6344 | 6720 | 5987 | 19046 | 48 | 45 | 38 | 133 |

GLaVE-Cap consistently produces significantly richer visual content. At the local caption level, it generates 30% more descriptive words compared to the ablated version. The advantage remains evident at the video-caption level (+18%), although some detail is inevitably compressed during summarization. These results demonstrate that the dual-stream structure enhances the model's ability to capture fine-grained visual information.

## K.2   OVERVIEW CAPTION ADDRESSES POOR CONTEXTUAL CONSISTENCY.

We compare ShareGPT4Video, GLaVE-Cap, and its ablated version that removes the overview caption injection, by asking GPT-4o to count the number of contextual inconsistencies across local captions and in the video captions. Inconsistencies include: (1) object inconsistency: a human/object transforming into an entirely different one; (2) attribute inconsistency: the same human/object abruptly changing in color, shape, etc., without any clear cause. Tab. 10 shows the average metrics across all 55 videos.

Table 10: Contextual inconsistency across local and video captions (per 1k words).

| Method | Local ↓ | Video ↓ |
|---|---|---|
| ShareGPT4Video | 0.375 | 0.429 |
| w/o Overview Caption | 0.145 | 0.086 |
| GLaVE-Cap | 0.102 | 0.046 |

Even the ablated version leads to significantly higher local-global consistency than ShareGPT4Video, demonstrating our TrackFusion design enables more detailed local captions that benefit the following summary stages in terms of ensuring contextual consistency. Incorporating the overview caption brings further improvement. We further manually inspect all video captions and 100 local captions. GPT-4o misidentified fewer than 5% of inconsistencies, suggesting the reliability of the LLM-based analysis. Notably, in our caption, inconsistencies rarely occur in key elements such as people and objects, since they are typically covered by the overview caption.

### K.3 Adaptive Scene-split addresses information loss during summarization.

We compare GLaVE-Cap and its ablated version that removes the adaptive scene-split module. Specifically, we ask GPT-4o to check whether each visual description in the local captions is preserved in the final video-level caption. We report the average of this proportion, calculated across all local captions from the 55 videos, as shown in Tab. 11.

Table 11: Proportion of preserved visual descriptions in final video captions.

| Method | Scene | Object | Person | Average |
|---|---|---|---|---|
| w/o Adaptive Scene-split | 72.09 | 57.33 | 58.61 | 62.89 |
| GLaVE-Cap | 90.80 | 81.80 | 77.07 | 83.77 |

Incorporating adaptive scene-split leads to a clear improvement in retaining detailed information (+20%). These results confirm that the adaptive scene-split effectively reduces information loss during summarization, thereby producing more comprehensive and faithful video captions.

## L   Impact of Prompt Design and Base Model Capability

### L.1   Analysis of how different types of prompts affect model performance

The prompt design plays a vital role in the performance of GLaVE-Cap. Our original prompts are based on a hierarchical structured design, inspired by prior works such as ShareGPT4Video and LLaVA-Video. We further investigate how different prompt templates affect the performance of the annotation model.

We focus on two subsets of the prompts used in GLaVE-Cap: those crucial for TrackFusion to extract visual details for local captions, and those crucial for CaptionBridge to summarize local captions into global captions. We choose two alternative prompt styles for evaluation:

- **Simplified Version:** Prompts retain only necessary parts to describe the task, input, and output, while removing specific guidance (e.g., constraints, skills, enumerated visual aspects).
- **Unstructured Version:** Prompts are rewritten by GPT-4o into a natural language paragraph without subtitles or lists, while preserving semantic content.

To assess the impact of different prompt types, we modify one prompt subset at a time and keep the remaining prompts unchanged. All experiments are conducted using GPT-4o as the annotation model. Due to API cost, we ran the experiment on a randomly sampled subset of *GLaVE-Bench*, which includes 10 videos and 1,090 questions. Accuracy results are shown in Tab. 12:

Table 12: Effect of different prompt styles on GLaVE-Cap performance.

| Version | Local Caption | Global Caption |
|---|---|---|
| Original Version | 69.85 | 69.85 |
| Simplified Version | 68.01 | 63.39 |
| Unstructured Version | 69.57 | 72.35 |

The results show that simplified prompts significantly degrade caption quality, suggesting that task-specific guidance is crucial for eliciting high-quality responses. Interestingly, unstructured prompts lead to similar or even better performance (+2.5 for global captions). This reveals that GLaVE-Cap may further benefit from prompt optimization strategies. The results also suggest that global caption generation is more sensitive to prompt design.

### L.2   Analysis of how the capabilities of (M)LLMs affect video captioning

To better understand how the capabilities of (M)LLMs affect video captioning, we conducted additional experiments. Beyond the results already reported for GPT-4o and Qwen2.5-VL-72B, we

further evaluated two smaller models, Qwen2.5-VL-32B and Qwen2.5-VL-7B, on *GLaVE-Bench*. Notably, Qwen2.5-VL-7B struggles to understand the prompt and fails to complete the task, often producing repetitive patterns and unreliable structured output. Qwen2.5-VL-32B performs relatively better but still shows limitations, such as failing to provide valid scene segmentation or generating captions in structured formats despite explicit instructions. Quantitative results are shown in Tab. 13:

Table 13: Performance of different (M)LLMs on *GLaVE-Bench*.

| Model | Acc ↑ | Hall ↓ | N.M. ↓ |
|---|---|---|---|
| GPT-4o | 67.68 | 16.04 | 19.40 |
| Qwen2.5-VL-72B | 63.67 | 17.42 | 22.90 |
| Qwen2.5-VL-32B | 61.05 | 18.74 | 24.87 |
| Qwen2.5-VL-7B | N/A | N/A | N/A |

We further explore the influence of the annotation model in individual modules of GLaVE-Cap, including TrackFusion and the summary step of CaptionBridge. We use GPT-4o as the main annotation model, but replace those modules with Qwen2.5-VL-32B. Results are shown in Tab. 14:

Table 14: Module-wise replacement analysis on *GLaVE-Bench*.

| Setting | Acc ↑ | Hall ↓ | N.M. ↓ |
|---|---|---|---|
| All GPT-4o | 67.68 | 16.04 | 19.40 |
| Qwen replace TrackFusion | 58.34 | 18.42 | 28.49 |
| Qwen replace Summary Step | 65.08 | 16.66 | 21.91 |

These results indicate that local caption generation (TrackFusion) relies more heavily on powerful annotation models, while the summarization step shows relatively weaker dependence.

## M  MORE ABLATION STUDIES

We have conducted ablation studies on both *VidCapBench* (Chen et al., 2025) and the short subset of *Video-MME* (Fu et al., 2025) (Video-MME-S). The settings are identical to Table 4, but use Qwen2.5-VL-72B as the annotation model due to the prohibitive cost of GPT-4o.

Table 15: Ablation study on *VidCapBench* and *Video-MME-S*.

| Method | VidCapBench-$Acc.$ | VidCapBench-$Pre.$ | VidCapBench-$Cov.$ | Video-MME-S |
|---|---|---|---|---|
| w/o Adaptive Scene-split | 16.52 | 56.50 | 88.69 | 65.11 |
| w/o Dual-stream Structure | 17.13 | 57.59 | 89.05 | 70.78 |
| w/o Overview Caption | 18.58 | 58.39 | 92.41 | 72.67 |
| w/o Visual Prompt | 18.96 | 58.57 | 92.49 | 73.00 |
| **Ours** | 18.78 | 60.98 | 92.34 | 73.22 |

The observed trends in Tab. 15 are generally consistent with those on GLaVE-Bench. However, the improvements are less pronounced, largely because *VidCapBench* consists mainly of short ($\approx$10s) clips, and *Video-MME* focuses on video reasoning rather than comprehensive and fine-grained video understanding.

Additionally, in Appendix K, we provide a GPT-4o-based quantitative analysis on object attribute and relation description, contextual consistency, and summarization quality, demonstrating the effectiveness of our proposed modules.

These results suggest that the modest gains on existing benchmarks stem from their limited capacity to capture the types of fine-grained improvements our method offers, highlighting the motivation and necessity for GLaVE-Bench.

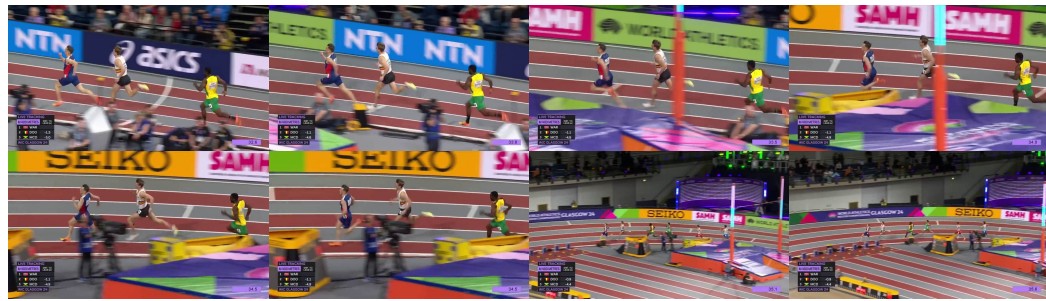

Three male athletes are captured mid-stride on a track, with the camera positioned to capture their movement from a side angle. **The athlete in the foreground, wearing a red and blue** uniform, appears to be leading the race. **The second athlete, dressed in white and black,** is slightly behind, while **the third athlete, in yellow and green,** is further back. The track is marked with lane lines, ...

As the scene progresses, the athletes continue their race on the track. **The athlete in the foreground, wearing a blue and red uniform, is now closer to the finish line** compared to the first frame, indicating progress in the race. **The athlete in the white and black uniform** has moved slightly forward, **maintaining a close competition with the leader. The athlete in the yellow and green** uniform remains in the lead, with **a slight increase in distance** from the others....

As the video played, the athletes continue their race on the indoor track, **with the camera angle providing a wider view of the track and the surrounding area**. **The athletes mentioned before** are steadily moving closer to the finish line, **maintaining their lead over the newly emerging athletes... The audience appears more engaged, with some standing and cheering, adding to the intensity of the race.**

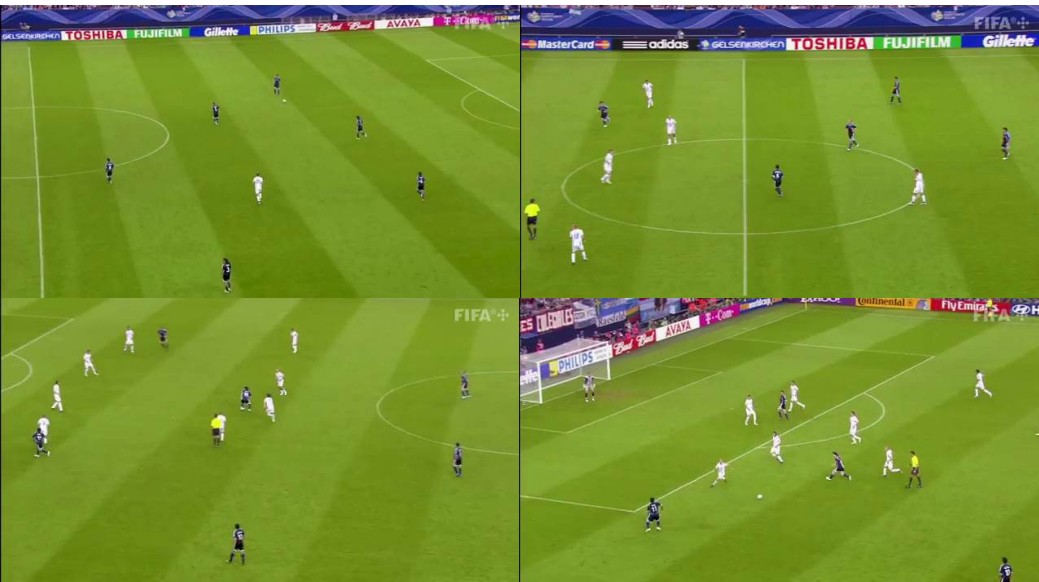

Initially, the focus was on **a player in a dark uniform dribbling the ball,** with a **player in a white uniform nearby, possibly preparing to intercept**...

The player who was previously dribbling the ball is no longer in the immediate foreground, indicating a shift in the play's focus. **The ball has moved slightly towards the center of the field,** prompting players from both teams to adjust their positions accordingly. **Players in dark jerseys,** including one near the bottom center of the frame and another closer to the center-right, are actively **engaged in the play. Meanwhile, players in white jerseys are strategically positioned,** with one near the center-left and another closer to the top-center area. **The referee, in a bright yellow uniform, is positioned centrally,** overseeing the gameplay and ensuring fair play...

As the match continues, the players dynamically reposition themselves, responding to the evolving gameplay.**A player in a dark jersey with the number 11 remains near the bottom left, positioning to receive the ball**, while another player in a dark jersey, **number 19 Messi, has shifted closer to the left, aligning with the ball's new location and making a pass to a teammate**. **Players in white jerseys adjust their positions, converging to set up their defense,** with one standing slightly further back and another repositioning slightly to the right. A new player in a white jersey enters the frame from the top-left corner, integrating into the ongoing gameplay..

Figure 11: Showcase two challenging sports-broadcast videos.

# N    SHOWCASE CAPTION

In this section, we showcase a full caption generated in GLaVE-Cap in Figs. 13 and 14.

We also showcase two challenging sports-broadcast videos with multiple objects moving simultaneously in GLaVE-Bench in Fig. 11. In the sprint example, the model reliably tracks rapid running athletes and provides precise descriptions for each athlete. In the soccer example, GLaVE-Cap remains stable and consistently captures the main events, such as passes and offensive intent, even if there are over ten people doing different actions.

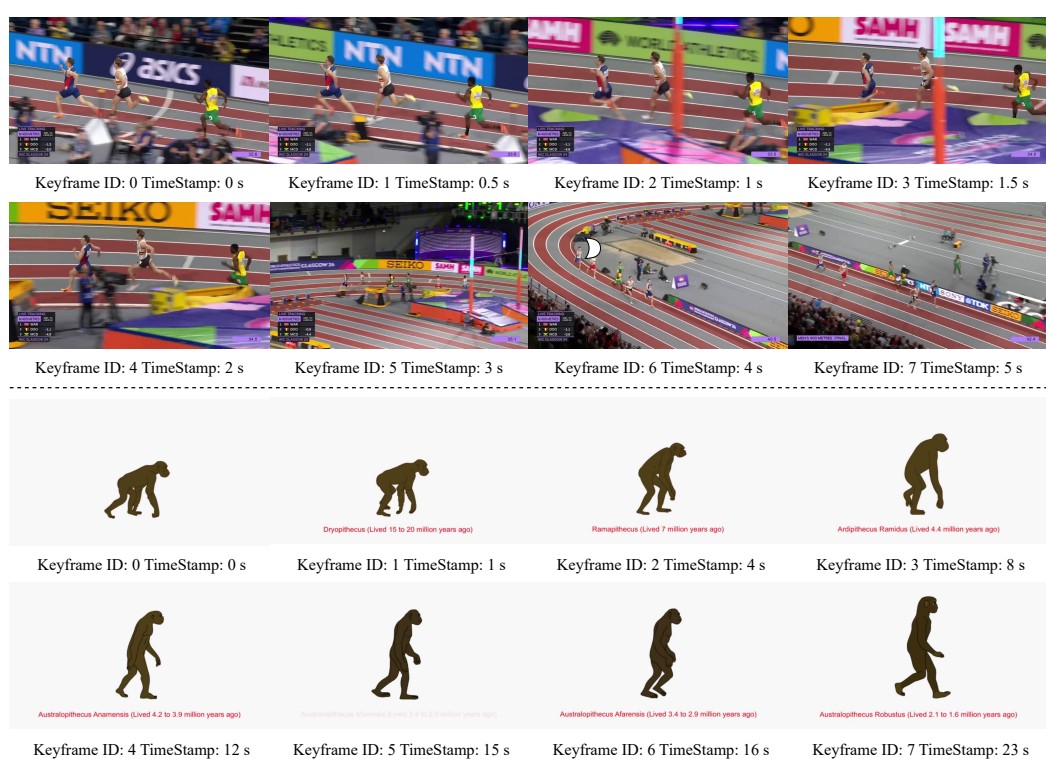

Figure 12: Showcase keyframe sampling for videos with varying temporal dynamics.

# O    SHOWCASE KEYFRAME SAMPLING FOR FAST-CHANGING VS. SLOW
## VIDEOS

To demonstrate the efficacy of our adaptive keyframe sampling strategy, we present a comparative showcase on videos with varying temporal dynamics, as illustrated in Fig. 12.

The upper portion features a sprint race (video 2yGaTOzaGIA in GLaVE-Bench). Here, even when the runner's core pose is stable, the drastically shifting background signals rapid motion, which triggers our CLIP sampler to adopt a dense sampling rate of one keyframe every 0.5 seconds to capture these essential dynamics. Conversely, the lower portion of Fig. 12 shows a segment from a slow-paced educational video (video HwnB8aCn8yE in GLaVE-Bench), featuring a mostly static content against a stable background. Here, the visual content remains largely consistent over time, allowing our sampler to select far fewer keyframes (about 4s per keyframe) while still preserving the semantic integrity of the narrative.

# P    GLAVE-CAP FRAMEWORK PROMPT TEMPLATE

In this section, we showcase the prompts used in GLaVE-Cap as Figs. 16 to 22.

## Q  BENCHMARK EVALUATION PROMPT TEMPLATE

We have introduced an automatic evaluation strategy for GLaVE-Bench in Sec. 4.1. The prompt template used is illustrated in Fig. 23.

## R  LIMITATION

Although our current framework leverages vision experts to assist the VLM in generating local captions, leading to clear improvements over using the VLM alone, it remains constrained by the VLM's capacity to process complex visual inputs within a single inference. In scenes containing numerous objects, the framework may still fail to capture important visual elements due to this bottleneck. To address this limitation, we plan to shift from keyframe-based to object-centric local captioning, enabling accurate object identification and comprehensive scene description through object tracking and multi-round interactions between vision experts and the VLM.

## S  LICENSE

We will introduce licenses of our assets and the sources, usage, and licenses of existing assets used in this work:

- **Code: Apache 2.0**
    - **PySceneDetect** v0.6.6: Used for scene detection. URL: `https://github.com/Breakthrough/PySceneDetect`. Licensed under the **BSD-3-Clause License**.
    - **Grounded-SAM2**: Used for object detection and segmentation. URL: `https://github.com/IDEA-Research/Grounded-Segment-Anything`. Licensed under the **Apache License 2.0**.
- **Test Benchmark: Video-MME License (Academic use only; no commercial use; no redistribution)**
    - We use the video from **Video-MME** (Fu et al., 2025). URL: `https://github.com/MME-Benchmarks/Video-MME`. The dataset follows the Video-MME license (Video-MME is only used for academic research. Commercial use in any form is prohibited. The copyright of all videos belongs to the video owners.).
- **Training dataset: Apache 2.0**
    - We use the video from **LLaVA-Video-Interleaved-178K** (Zhang et al., 2024). URL: `https://huggingface.co/datasets/lmms-lab/LLaVA-Video-178K`. The dataset follows the **Apache License 2.0** license. We respect the license terms and provide attribution accordingly.
- **Pretrained Model: Apache 2.0**
    - We fine-tuned our model based on **Qwen2.5-VL-7B** (Bai et al., 2025), developed by Alibaba. URL: `https://github.com/QwenLM/Qwen2.5-VL`. The model is licensed under the **Apache License 2.0**. We comply with its terms for research use.

## T  USE OF LARGE LANGUAGE MODELS

In the preparation of this manuscript, the author utilized a Large Language Model (LLM) exclusively for post-writing assistance to refine language and clarity. The LLM was not used to generate ideas or content but solely for tasks such as grammar correction and sentence rephrasing. The author thoroughly reviewed and takes full responsibility for all output.

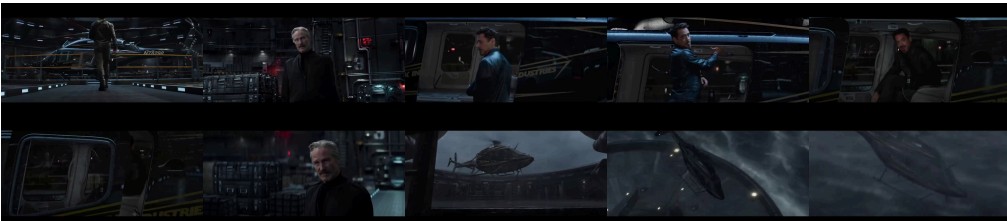

The video begins in a high-tech hangar, where a sleek, modern helicopter bearing the "Stark Industries" logo is prominently displayed. The helicopter, painted in a glossy dark color, is positioned centrally in the frame, surrounded by safety railings and illuminated by dim, ambient lighting that casts soft shadows, enhancing the dramatic atmosphere. A man in a dark jumpsuit, suggestive of a pilot or technician, is seen walking purposefully towards the helicopter, indicating a sense of readiness and anticipation for an upcoming event or mission. The hangar itself is spacious and industrial, with metallic surfaces, high ceilings, and numerous lights and beams contributing to the futuristic ambiance. As the scene progresses, another individual enters from the right, partially visible and dressed in light-colored pants, suggesting their involvement in the forthcoming action. The helicopter remains the central focus, with its rotor blades silhouetted against the well-lit ceiling, maintaining the controlled and secure environment of the hangar. The scene then shifts focus, with the helicopter and the man walking towards it no longer visible. Instead, the attention is on the individual in the foreground, dressed in a dark outfit, facing the viewer in a moment of pause or contemplation. To the left, a large, sturdy metal container with reinforced edges and multiple latches is prominently displayed, marked with various symbols and a numeric identifier "10." The background reveals a different part of the hangar, with structural elements like ladders, industrial piping, and illuminated panels, adding to the industrial feel. The subdued lighting, with a few red lights glowing, suggests operational status or security measures. The scene concludes with a medium shot capturing both the person and the container, allowing viewers to appreciate the scale and environment of the hangar, setting the stage for the narrative's impending action or departure.

As the scene progresses, the focus shifts from the high-tech hangar to the exterior environment, where a Stark Industries helicopter is prominently featured. Initially, a man in a dark leather jacket is seen standing near the helicopter, suggesting an imminent interaction with the aircraft. The helicopter, marked with the "Stark Industries" logo, is sleek and modern, with its dark exterior reflecting the dim lighting and raindrops, indicating rainy weather. The man, with a purposeful stance, appears ready to engage with the helicopter, possibly preparing for a significant event or mission. As the narrative unfolds, the man is seen interacting with the helicopter's open door, suggesting he is about to enter. The environment remains consistent with the high-tech and dramatic atmosphere, enhanced by the subdued lighting and the presence of rain. The camera captures the scene from various angles, emphasizing the interaction between the man and the helicopter, and highlighting the advanced technology associated with Stark Industries. The scene then transitions to the interior of the helicopter, where the man is now seated, indicating he has entered the aircraft. The helicopter's interior is dimly lit, with visible details such as seats and control panels, reinforcing the high-tech setting. The presence of rain on the helicopter's exterior continues to underscore the rainy weather conditions. The camera provides a medium shot, focusing on the man and the helicopter's interior, suggesting readiness for departure. In the final frames, the man is no longer visible inside the helicopter, leaving the space he occupied empty. The helicopter remains stationary, with its open door suggesting imminent activity. The scene captures the helicopter's sleek design and the rainy environment, maintaining the suspense and anticipation of a significant event or departure. The overall composition emphasizes the advanced technology and the high-stakes atmosphere, aligning with the narrative of a pivotal mission or event about to unfold.

As the video scene progresses, the focus remains within a high-tech hangar, which exudes a controlled and sophisticated atmosphere accentuated by dim lighting and advanced industrial elements. Initially, the Stark Industries helicopter remains a significant presence, partly visible with reflections shimmering off its dark, sleek surface. This reflection, along with dim lighting, contributes to a sense of control and anticipation as the scene unfolds. Attention then shifts within the hangar to an individual standing purposefully in front of an array of large, stacked, dark-colored crates. These crates, secured with metal clasps, suggest the presence of valuable or sensitive equipment. The person, dressed in a stylish dark coat, carries an authoritative demeanor and is positioned slightly to the right of the scene's center, facing the left side. Their posture suggests observation or potential interaction with the crates, possibly indicating preparation or readiness for forthcoming events. The backdrop provides a broader view of the hangar's interior, revealing metallic structures and equipment typical of a high-tech, industrial setting. The presence of faint red lights punctuates the atmosphere, perhaps hinting at operational status or security measures. The floor below displays a grid-like pattern, contributing to the industrial environment's emphasis on durability. The scene transitions to a broader perspective, shifting the narrative focus beyond the hangar's interior to its exterior. Here, the Stark Industries helicopter once more becomes central, depicted in sharp relief against the darker, stormy surroundings. Positioned slightly off-center, its streamlined design, water reflecting off its surface, and the expansive, curved architecture of the hangar's exterior together frame a sense of imminent departure. The wide-angle camera perspective captures this anticipation, highlighting both the helicopter's technological prowess and its role in a significant upcoming event, as foreshadowed by the overview caption.

As the scene progresses, the focus remains on the individual inside the Stark Industries helicopter, which is in flight amid stormy weather. The helicopter's interior is dimly lit, creating a tense and somber atmosphere that aligns with the narrative of a high-stakes mission. The person, wearing a dark leather jacket, is seated and appears to be preparing for a significant task. They are seen interacting with an object, likely the advanced technology related to the Iron Man suit, as suggested by the overview caption. The helicopter's modern and sleek design is evident, with plush light-grey leather seats and a neutral gray color scheme that enhances the high-tech ambiance. The environment outside the helicopter is characterized by dark, stormy skies and rain, visible through the large window, which adds to the dramatic and adventurous feel of the scene. The rain-soaked glass and the turbulent weather conditions underscore the urgency and intensity of the mission. The camera angle provides a close-up view of the individual and their actions, centralizing them as the focal point amidst the advanced, tech-driven setting. Throughout the scene, the individual makes subtle movements, such as adjusting their posture or interacting with the object in their hands, indicating a state of readiness and anticipation. The presence of safety labels, including a no-smoking sign, reinforces the controlled and professional environment within the helicopter. The overall composition of the scene, with its dim lighting and stormy backdrop, creates a sense of impending action and highlights the individual's preparation for a critical moment involving the Iron Man suit.

Figure 13: The first part of the showcase caption

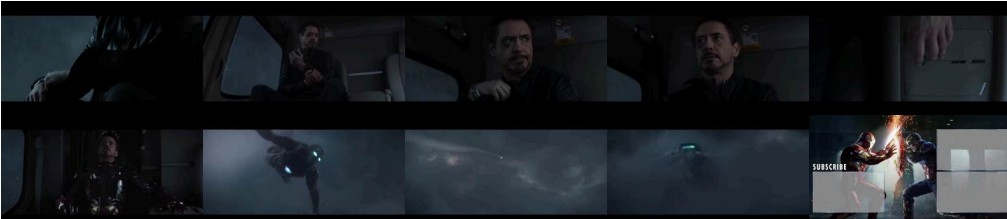

As the scene unfolds, the focus is on the interior of a helicopter, where a hand, initially at rest, engages with a compartment latch. The hand, with a light skin tone, grips the metallic latch, attempting to open the compartment. The compartment is sleek, with metallic accents and the text "LIFT" and "HANDLE," along with "STARK INDUSTRIES" branding. The dim lighting emphasizes the hand's action, creating a focused ambiance. As the scene progresses, the hand, now partially clad in a red and silver metallic gauntlet, interacts with the compartment, revealing a red cylindrical object, likely part of the Iron Man suit. The compartment opens further, showcasing more of its contents, with the lighting highlighting the high-tech nature of the gauntlet. The perspective shifts to a broader view within the helicopter, revealing a person in a partially assembled Iron Man suit. The suit's red and gold colors are prominent, with the individual seated centrally, arms extended, suggesting the process of donning the armor. The dimly lit interior enhances the suit's glossy surface, with a window showing a stormy backdrop. The camera captures the upper body of the individual, emphasizing the high-tech appearance of the suit and the character's preparation for a mission. The scene transitions to the individual fully equipped in the Iron Man suit, with the glowing blue arc reactor on the chest illuminating the area. The suit is depicted in a dynamic posture, indicating readiness for action. The helicopter's interior remains dark, with the blue glow from the suit creating a futuristic ambiance. The framing captures Iron Man's commanding presence, hinting at his mission-ready state. A dramatic shift occurs as the scene moves from the helicopter's interior to an outdoor setting. Initially, Iron Man stands near a doorway, illuminated by the arc reactor. The subsequent frame captures the helicopter in mid-flight amidst stormy skies, with Iron Man absent, indicating his departure into the storm. The helicopter's dark exterior and bright navigation lights enhance the dramatic atmosphere. Iron Man is then depicted flying through the stormy sky, his suit's glowing elements contrasting against the dark clouds. Positioned diagonally, his forward motion is emphasized, conveying power and purpose. The helicopter is no longer visible, highlighting Iron Man's transition to independent flight. The scene's lighting remains dim, maintaining a moody atmosphere. As the scene progresses, Iron Man vanishes from view, leaving only the stormy sky. The focus shifts to the ambient environment, with swirling clouds dominating the scene. A bright, glowing object, likely Iron Man, appears, cutting through the clouds and emphasizing dynamic action. The camera captures the vastness of the sky, with Iron Man's presence providing a focal point. Finally, Iron Man reappears, flying dynamically through the stormy atmosphere. His illuminated suit contrasts with the dark sky, emphasizing his presence and readiness. The camera focuses on Iron Man's flight, capturing the dynamic movement and action-packed nature of the scene. The stormy environment remains intense, with Iron Man's flight underscoring his solitary strength and capability amidst the turbulent weather.

As the scene progresses, the focus is on the stormy sky, filled with thick, swirling clouds that create a moody and intense atmosphere. Initially, a small, indistinct object, likely a helicopter, is faintly visible in the lower right portion of the frame, silhouetted against the dark clouds. The helicopter's silhouette is partially obscured, adding to the mysterious and ominous ambiance. As the scene unfolds, the helicopter moves significantly upward and to the right, suggesting rapid ascent or navigation through the challenging weather conditions. The wide-angle camera perspective captures the vastness of the sky, emphasizing the intensity of the storm and conveying a sense of urgency and adventure. In the subsequent frames, the scene undergoes a transformation as the previously indistinct object disappears, replaced by a large, futuristic helicopter that emerges prominently. This helicopter, characterized by its sleek, angular contours and high-tech appearance, aligns with the distinctive style of Stark Industries machinery. Its upper section is adorned with bright teal-lit panels, likely serving as navigation lights or indicators, which stand out against the muted tones of the stormy surroundings. These lights illuminate the top of the craft slightly, emphasizing its aerodynamic design and advanced build, suitable for navigating the turbulent storm clouds. The camera angle, positioned slightly below the helicopter, enhances the sense of motion and elevation as the craft appears to be climbing or banking to navigate the weather. The surrounding environment is dominated by shades of grey and blue, portraying layers of thick mist and storm clouds that obscure detailed views of the landscape. The lighting and environmental conditions remain consistent with the dark and cloudy setting, emphasizing the helicopter's presence against the murky backdrop. This introduction of the helicopter suggests a pivotal moment, as it appears prominently and distinctly in the scene amidst the stormy skies, evoking a blend of mystery and tension.

As the scene progresses, the focus is initially on a helicopter flying through a dark, stormy sky. The helicopter, characterized by its sleek design and illuminated elements on its top, suggests advanced technology, consistent with Stark Industries' machinery. The environment is dominated by a moody, overcast atmosphere, with heavy clouds surrounding the aircraft, indicating inclement weather conditions. This setting aligns with the narrative of a high-stakes mission amid stormy skies. As the scene unfolds, the helicopter disappears from view, and the frame is enveloped in a dense, dark fog, creating a mysterious and somewhat ominous atmosphere. The visibility is significantly reduced due to the thick mist, casting a bluish-gray hue over the scene. This foggy environment suggests a setting that is either high in altitude or experiencing severe weather, such as a storm or heavy rain. The lighting is dim, contributing to the overall mood of uncertainty and suspense. The camera perspective is a wide-angle shot, capturing the vastness of the foggy environment without focusing on any particular subject. The disappearance of the helicopter emphasizes the dynamic and unpredictable nature of the stormy setting, possibly preceding a significant event or action. As the scene transitions, the fog and any potential light sources disappear, leaving the frame entirely black. This dramatic cut to black suggests a possible transition to a different scene or a significant shift in the narrative, aligning with the overview caption's mention of a confrontation between Iron Man and Captain America. The absence of any visible elements in the final frames emphasizes the stark contrast from the previous foggy environment, creating a sense of anticipation for the upcoming events.

As the scene commences, it opens with a stark black screen, setting an anticipatory tone that underscores a significant transition within the narrative. Suddenly, the darkness gives way to a vivid and intense confrontation between Iron Man and Captain America. Positioned on the left of the frame, Iron Man stands boldly in his red and gold armor, characteristic of Stark Industries' cutting-edge technology. The suit radiates under the focused lighting, highlighting his formidable presence and the advanced design typical of his armor. Iron Man assumes an aggressive stance, his right arm extended forward, hand open and bracing an energy projection directed towards Captain America. Opposite Iron Man, on the right, Captain America, clad in his patriotic blue, red, and white ensemble, stands resolute with his iconic shield raised defensively. The bright sparks that erupt where Iron Man's energy impacts the shield accentuate the ferocity of their clash, casting vivid bursts that highlight the confrontation's dynamic intensity. The dramatic contrast between the illuminated characters and the dark, indistinct background draws the viewer's attention squarely to the intense conflict. This lighting effectively captures the intricate details of their costumes, emphasizing the stark divergence of their allegiances and ethos. As the scene unfolds, Iron Man remains intensely focused on breaching Captain America's defenses. Throughout their struggle, sparks continuously fly from the impact zone, creating a vibrant visual metaphor for their clashing ideologies. Despite Iron Man's powerful assault, Captain America stands unyielding, epitomizing his enduring spirit and the strength embodied by his shield. The camera maintains an eye-level perspective throughout, ensuring that the gravity of their conflict remains visceral and immediate for the viewer. Notably, the word "SUBSCRIBE" appears in bold white text in the upper left corner of the frame during their confrontation. This addition serves as a call-to-action, embedded within the charged visual narrative. The enduring and intense focus on the rivalry between Iron Man and Captain America enhances the entire scene's captivating and suspenseful nature, leaving the audience engrossed in the unfolding drama.

Figure 14: The second part of the showcase caption

**ShareGPT4Video**
In the grocery store aisle, the scene has expanded with more people present. The man with the cap, who was previously holding the young child, is now kneeling down, supporting the child as they reach out towards the young girl. The girl in the light-colored dress with long dark hair is still present and appears to be interacting with the child. Another adult, a woman, stands nearby holding the child's hand, assisting in the interaction. The shelves, stocked heavily with various pantry items, remain consistent, while additional aisles are visible, emphasizing a larger view of the grocery setting.

**Ours**
The scene opens in a grocery store aisle, where Jyoti (ID=4), known as the world's smallest woman, is seen amidst a lively interaction. Dressed in a long yellow outfit, she stands on the right side of the aisle, interacting with an infant (ID=2) in a blue dress and light-colored shoes, who holds the hand of a woman (ID=7) dressed in a red top. ... To Jyoti's left, a man (ID=3) is kneeling, wearing a gray hoodie and a dark cap, maintaining his position in the scene. Beside him, on the ground, another man (ID=1) dressed in a blue hoodie and jeans remains engaged in conversation, slightly repositioned yet still kneeling. Further enhancing the scene's dynamic, a new person (ID=10) is visible on the left. Wearing a blue shirt and beige pants, he ... Above the aisle, a sign reads "Fresh DAIRY," and a "TLC" logo in the top-right corner indicates ... Meanwhile, colorful and vibrant packaging fills the shelves, making the setting inviting. The ambiance is cheerful as suggested by the caption "Baby, hi," reflecting ..., with the camera capturing all the activities from a close frontal angle.

Figure 15: The full caption in Fig. 1. We omit from our captions the parts not mentioned in Fig. 1, primarily those describing actions, reasoning, and overall stylistic elements.

**# Character**

You are an expert video frame analyst with exceptional attention to detail. Your task is to provide clear, fluent, and sequential descriptions of adjacent video frames. You will focus on identifying and explaining any changes between two adjacent frames, including alterations in object actions, behaviors, appearances, relationships, and environmental or camera movements. Supplementary JSON data detailing object bounding box positions will be provided for reference. The overview caption of the video will also be provided to improve overall contextual understanding.

**# Input Contents**

- Video Frames: Two adjacent frames in a video will be provided. For each frame, some specific objects of interest have been highlighted by colorful borders and unique numeric labels. The numeric label of the same object is consistent between the two frames.
- JSON Data: For each frame, a JSON data describing the bounding box positions of those objects of interest will be provided. The JSON format is as follows: `{<id>: {x1: <x1>, y1: <y1>, x2: <x2>, y2: <y2>}}`, where `<id>` is the numeric label of the object in the frame. This information can assist your analysis, including tasks such as counting objects, understanding spatial relationships, and ensuring no important objects are missed.
- Overview Caption: To provide context on the events in the video, an overview caption will be provided to summarize the scene or event along with its frame range [XX - XX], thus minimizing potential errors in intent recognition and enhancing overall contextual comprehension.

**# Skills**

**## Skill 1: Thorough Analysis of Objects**

Based on the given frames, develop a detailed understanding of objects, especially the changes between frames. Analyze the following aspects:
- Visual attributes: size, shape, color, quantity, recognizable text, any unique features, etc.
- Actions and behaviors: object movements, interactions, etc.
- Spatial locations: Use precise positional and relational terms (e.g., above, to the left, on the right, etc.)
- Relationships: interactions, dependencies, etc.

**## Skill 2: Capturing Environment and Camera Movements**

You should get a understanding of the following aspects, especially the changes between frames:
- Environment: scene setting, background details, lighting, recognizable text, etc.
- Camera movements: panning, tilting, zooming in, etc.

**# Tasks**

Your task is to thoroughly analyze the given frames. \*\*Provide a detailed description of changes between frames, focusing on variation of objects, environment, and camera dynamics.\*\* Emphasize information in the later frame by highlighting newly added objects, missing objects, and significant changes while maintaining an accurate comparison with the earlier frame. Pay special attention to the objects highlighted with colorful-borders and numeric labels, but also incorporate any other relevant changes happening in the scene.

**# Constraints**

- Stick to a narrative format for descriptions, avoiding list-like itemizations.
- Base your descriptions solely on visible evidence, avoiding analysis or speculation.
- Ensure descriptions are concise, fluent, and objective, avoiding rhetorical devices or irrelevant commentary.
- When provided with only one frame (the first frame), describe it directly without considering temporal connections.
- Do not explicitly mention frame numbers or timestamps.
- Exclude references to colorful boundaries and bounding box coordinates in your descriptions. They are meant to assist you, but your description should NOT include them.
- Use previous caption only to support your understanding of the context. Do not repeat or rephrase the provided previous caption.
- Whenever you mention a labelled object, always include its corresponding numeric ID in parentheses in the format: (ID=<id>). For example, "a man in a blue shirt and light pants (ID=5)" or "the young girl (ID=3)". Avoid referring to objects solely by their numeric IDs.

**# Structured Input**

Current Frame ID: `<ID>`
Overview caption: `<overview_caption>`
First Video frame: `<video_frame>`
First JSON: `<JSON>`
Second Video frame: `<video_frame>`
Second JSON: `<JSON>`

Figure 16: `prompt_diff`

**# Character**

You are an exceptional image analyst known for your sharp attention to detail. Your expertise lies in providing clear descriptions of images. I will provide you with an image. Your task is to analyze the image, focusing on identifying the main objects and their attributes, such as color, location, shape, and other important details. An edited version of the image, and supplementary JSON data detailing object bounding box positions will be provided for reference. The overview caption of the video will also be provided to improve overall contextual understanding.

**# Task Overview**

You will analyze a provided image and generate a **detailed, fluent description** that captures:
- **Main objects and their attributes**, including **size, shape, color, quantity, and unique features**.
- **Actions and interactions** between objects.
- **Spatial positioning** using precise relational terms (e.g., left, right, above, behind).
- **Scene and environment**, covering **background details, lighting, and recognizable text**.
- **Camera perspective**, such as **close-up, wide-angle, or overhead shots**.
- **All Recognizable text** in the image, such as subtitle, logo text, etc.

**# Input Contents**
- **Image**: The image to analyze.
- **Edited Image**: The edited version of the image, where some specific objects of interest have been highlighted by colorful borders and unique numeric labels.
- **JSON Data** A JSON data with bounding box coordinates for these labeled objects: {<id>: {x1: <x1>, y1: <y1>, x2: <x2>, y2: <y2>}}. This data helps you identify your analysis, including tasks such as counting objects, understanding spatial relationships, and ensuring no important objects are missed.
- **Overview caption**: To provide context on the events in the video, an overview caption will be provided to summarize the scene or event along with its frame range [XX - XX], thus minimizing potential errors in intent recognition and enhancing overall contextual comprehension.

**Note that** the two images are nearly identical. Analyze them as a single image, leveraging the edited version to better understand object boundaries.

**# Guidelines**

## **1. Object Identification and Analysis**
- Pay special attention to the objects highlighted with colorful-borders and numeric labels, but also incorporate any other relevant objects in the image.
- Whenever you mention a labelled object, always include its corresponding numeric ID in parentheses in the format: (ID=<id>). For example, "a man in a blue shirt and light pants (ID=5)" or "the young girl (ID=3)". Avoid referring to objects solely by their numeric IDs (e.g., "ID=5 is standing").

## **2. Language and Style**
- **Use a natural narrative style** rather than a list format.
- **Retaining all available information.**
- **Remain objective**—do not infer or speculate beyond visible evidence.
- **Exclude references to colorful boundaries or bounding box coordinates** in your descriptions. They are meant to assist you, but your description should NOT include them.

**# Structured Input**
Current Frame ID: `<ID>`
Overview caption: `<overview_caption>`
Image: `<Image>`
Edited Image: `<Edited Image>`
JSON: `<JSON>`

Figure 17: $\texttt{prompt}_{\texttt{detail}}$

**# Character**

You are an expert video frame analyst with exceptional attention to detail. Your task is to generate clear, fluent, and sequential descriptions of changes between adjacent video frames. The overview caption of the video will also be provided to improve overall contextual understanding.

**# Task Overview**

You will be provided with:
- **Overview caption**: To provide context on the events in the video, an overview caption will be provided to summarize the scene or event along with its frame range [XX - XX], thus minimizing potential errors in intent recognition and enhancing overall contextual comprehension.
- **Differential Caption**: A detailed description of changes between two adjacent frames, focusing on variations in objects, environment, and camera dynamics.
- **Supplementary Material**: A detailed description of the later frame, providing additional static attributes and background details.

Note: in the provided contents when mentioning specific objects, its unique ID will follow in parentheses. The ID of the same object is always consistent. You can refer to the ID to better assist your analysis.

Your task is to generate a **more accurate, detailed, and comprehensive differential caption**, by expanding on the main subjects, their actions, and the background using the provided supplementary material.

**# Guidelines**

**## **1. Prioritization in Case of Conflicts****
- **Object actions and movements**: Follow the **Differential Caption** (since it emphasizes changes).
- **Object static attributes (size, color, position, etc.)** and ANY other description: Follow the **Supplementary Material** (since it provides an accurate and thorough description).
- **Background details**: Follow the **Supplementary Material**, unless the **Differential Caption** indicates a change.

In short, you need to retain the changes described in the differential caption and replace any vague or incorrect details in the differential caption with the descriptions provided in the supplementary material.

**## **2. Key Aspects to Cover****

Retain as much information as possible, even if it appears in only one description or supplementary detail. Ensure descriptions accurately reflect:
- **Visual attributes**: Object appearance, color, size, shape, unique features, recognizable text, etc.
- **Actions and interactions**: Movement, manipulation, relationships between objects, etc.
- **Spatial locations**: Precise positional and relational terms (e.g., left, right, above, behind).
- **Environmental context**: Scene setting, background details, lighting, recognizable text.
- **Camera movement**: Changes in angle, zoom, panning, stabilization, etc.
- **Recognizable text** in the video.

**## **3. Language and Style****
- **Use a natural narrative style** rather than a list format.
- **Retaining all available information** in both differential caption and supplementary material, especially the objects with ID and relative description. All labeled objects and their corresponding descriptions in the supplementary material must be included.
- **Remain objective**—do not infer or speculate beyond the provided data.
- Whenever you mention a labelled object, always include its corresponding unique ID in parentheses in the format: (ID=<id>). For example, "A red cup (ID=7) on the table is picked up by the man (ID=5) and tilted forward, pouring out liquid." Avoid referring to objects solely by their numeric IDs.

**# Structured Input**

Current Frame ID: `<ID>`
Overview caption: `<overview_caption>`
Differential Caption: `<caption>`
Supplementary Material: `<material>`

Figure 18: prompt_merge

# Task Description
You will receive keyframes from a video and need to generate a concise yet accurate description based on these keyframes. Your description should summarize the main content of the video without excessive details.

# Guidelines For Description
- Provide a concise summary, avoiding unnecessary details while ensuring key information is preserved.
- Include the following essential elements:
    - Scene and background information (environment, time, location, weather, objects, etc.).
    - Character information (identity, relationships, emotions, intentions).
    - Event logic and timeline (causes, current state, subsequent developments).
    - Interaction and causality (interactions between people or objects, causal relationships of actions).
    - Key visual elements (facial expressions, posture, gestures, body orientation, etc.).
    - Keyframe range (to help analyze and understand the context).

Important Notes:
- Each sentence **MUST** be accompanied by a keyframe range [XX - XX] to help track context across multiple scenes. If necessary, the keyframe range can also be included within the sentence itself.
- Ensure the description is concise yet informative, avoiding unnecessary details while capturing key elements.
- When describing interaction and causality, focus on logical relationships rather than isolated actions.

Figure 19: `prompt_overview`

# Character
You are an advanced video analysis assistant. I have a video composed of a series of frames, numbered sequentially from 1 to n. I will provide you with **detailed captions**, the i-th **detailed caption** explains the detail description of the i-th frame. Additionally, I will provide you with an automatic scene segmentation result based on shot transitions in JSON format (e.g.,{"0":[0,7],"1":[8,17],...}). You can understand what happens in each frame, and get a coherent understanding of the video timeline.

# Task
Split the video into clips, ensuring that each clip focuses on one primary topic, event, or background. Utilize the automatic scene segmentation to identify background shifts and refer to the detailed captions to grasp the topic and event. Finally, generate the most optimal scene segmentation result.

# Constraints
For each event, provide:
- Frame range: The starting and ending frame numbers for the event.
- Scene_hint: Used to refer to the scene, ensuring that it uniquely and accurately identifies the scene. Each scene hint should also be kept as concise as possible to avoid disclosing excessive scene details.
- Camera: A description of the camera movements or techniques used, if applicable.

# Output Format
The output should be a valid JSON structure with the following format:
```
{
  "0": {
    "frame": [1, 12],
    "scene_hint": "Display of Motorcycle Details",
    "camera": "Camera pans across the motorcycle and occasionally zooms in for detail shots."
  },
  "1": {
    "frame": [13, 28],
    "scene_hint": "Journey Through the Countryside",
    "camera": "The camera tracks the motorcycle's motion with occasional aerial views."
  }
}
```

# Notes
- Only output the JSON structure; DO NOT include any additional explanations or commentary. Ensure the output JSON is syntactically correct and follows the example structure.
- Do not separate transition frames into a distinct scene; they should be included in the next scene.

# Structured Input
The 1-th detailed caption: `<caption_1>`
The 2-th detailed caption: `<caption_2>`
...
The n-th detailed caption: `<caption_n>`
Automatic scene segmentation result: `<segmentation_result>`

Figure 20: `prompt_ss`

**# Task Overview**

You are an advanced video analysis assistant. There is a video including multiple sequential scenes(scene-1,scene-2,XXX). Now I have a video scene composed of a series of frames, numbered sequentially from 1 to n. You will be provided with:
- **Overview caption**: To provide context on the events in the video, an overview caption will be provided to summarize the scene or event along with its frame range [XX - XX], thus minimizing potential errors in intent recognition and enhancing overall contextual comprehension.
- **Previous Scene Descriptions**: The detailed descriptions describing multiple earlier video scenes.
- **Transition Captions**: Transition captions describe the transitions between consecutive frames. The i-th **transition caption** explains the changes from the (i-1)-th frame to the i-th frame, typically incorporating details about the (i-1)-th frame as well as the changes that occurred. **Note: These captions describe transitions, not solely the content of individual frames.**

Using the overview caption and previous scene descriptions as context, you are required to create the descriptions for the current scene based on the transition captions provided.

**# Guidelines For Scene Description**
- Your description should see the previous scene descriptions as context.
- Analyze the narrative progression implied by the sequence of frames, interpreting the sequence as a whole.
- Note that since these frames are extracted from a clip, adjacent frames may show minimal differences. These should not be interpreted as special effects in the clip.
- If text appears in the frames, you must describe the text in its original language and provide an English translation in parentheses. For example: 书本 (book).
- When referring to people, use their characteristics, such as location, clothing, to distinguish different people.
- When referring to an entity that appeared in the previous scene, explicitly indicate its continuity at the beginning of the scene (e.g., "the man in the previous scene" or simply "the man" to imply the connection). Avoid excessive repetition when describing entities that have already been introduced.
- Summarize the progression of the scene and the actions of the humans, without describing the changes for each individual frame transition.
- Ensure the output is clear and objective, relying solely on explicitly stated details in the transition captions without speculation or inference. Avoid referencing timestamps or frame indexes.
- **IMPORTANT** Please provide as many details as possible in your description, including colors, shapes, and textures of objects, actions and characteristics of humans, spatial relationships among people and objects, camera movements and transitions, as well as the overall scenes and backgrounds. Note that DO NOT repeat details that have already been described if they remain unchanged.

**# Input Format**
Current Scene Frame Range: [<START_ID> - <END_ID>]
Overview caption: <overview_caption>
Previous Scene Descriptions:
<previous_scene_descriptions>

Transition caption between <timestamp_0> seconds and <timestamp_1> seconds: <caption_1>
Transition caption between <timestamp_1> seconds and <timestamp_2> seconds: <caption_2>
...
Transition caption between <timestamp_n-1> seconds and <timestamp_n> seconds: <caption_n>

**# Output Format**
1. Only provide the string which describes the scene.
2. You can use various descriptive sentence structures to outline the narrative progression. One example is: "As the video progresses,... As the scene progresses...".

Figure 21: $\text{prompt}_{\text{sc}}$ for the non-initial scene.

**# Task Overview**

You are an advanced video analysis assistant. I have a video scene composed of a series of frames, numbered sequentially from 1 to n. You will be provided with:
- **Transition Captions**: Transition captions describe the transitions between consecutive frames. The i-th **transition caption** explains the changes from the (i-1)-th frame to the i-th frame, typically incorporating details about the (i-1)-th frame as well as the changes that occurred. **Note: The first transition caption only describes the first frame. Other transition captions describe transitions, not solely the content of individual frames.**

You are required to create the descriptions for the scene based on the transition captions provided.

**# Guidelines For Scene Description:**
- Analyze the narrative progression implied by the sequence of frames, interpreting the sequence as a whole.
- Note that since these frames are extracted from a clip, adjacent frames may show minimal differences. These should not be interpreted as special effects in the clip.
- If text appears in the frames, you must describe the text in its original language and provide an English translation in parentheses. For example: 书本 (book).
- When referring to people, use their characteristics, such as location, clothing, to distinguish different people.
- Summarize the progression of the scene and the actions of the humans, without describing the changes for each individual frame transition.
- Ensure the output is clear and objective, relying solely on explicitly stated details in the transition captions without speculation or inference. Avoid referencing timestamps or frame indexes.
- **IMPORTANT** Please provide as many details as possible in your description, including colors, shapes, and textures of objects, actions and characteristics of humans, spatial relationships among people and objects, camera movements and transitions, as well as the overall scenes and backgrounds.

**# Input Format**

Caption describing `<timestamp_1>` seconds: `<caption_1>`

Transition caption between `<timestamp_1>` seconds and `<timestamp_2>` seconds: `<caption_2>`

...

Transition caption between `<timestamp_n-1>` seconds and `<timestamp_n>` seconds: `<caption_n>`

**# Output Format**

1. Only provide the string which describes the scene.

2. You can use various descriptive sentence structures to outline the narrative progression. One example is: "The video begins with... As the scene progresses... The scene concludes with...".

Figure 22: $\texttt{prompt}_{\texttt{SC}}$ for the first scene.

**# Task**

You will be given a detailed description of a video along with a brief description of one specific scene from the video. Your task is to:

1. Comprehend the overall content of the video based on the provided caption.

2. Identify the specific scene referred to in the given scene description. (To assist you, descriptions of the scenes immediately before and after will also be provided.)

3. Answer a multiple-choice question related to the identified scene.

**# Constraints**

-. If the given scene description does not provide enough details to confidently identify a specific scene, or if the caption lacks sufficient information to answer the question, choose the implicit option: "E. Not enough information mentioned".

-. Begin by explaining your reasoning—how you located the scene and arrived at your answer.

-. Finally, output your answer in the form of a Python dictionary string with the key 'answer' and a value of either 'A', 'B', 'C', 'D', or 'E', such as: {'answer': 'B'}.

**# Structured Input**

[caption] `<full video description>`

[pre-scene] `<Description of the scene immediately before>`

[scene] `<Brief description of the target scene>`

[post-scene] `<Description of the scene immediately after>`

[question] `<question>`

[options]

`<choice A>`

`<choice B>`

`<choice C>`

`<choice D>`

Figure 23: Prompt for GLaVE-Bench evaluation

