# OpenReview forum: "GLaVE-Cap: Global-Local Aligned Video Captioning with Vision Expert Integration"
_ICLR.cc/2026/Conference — Submitted to ICLR 2026_

### Official Review · Reviewer_PK8P · 2025-10-27

**Soundness:** 2
**Presentation:** 3
**Contribution:** 3
**Rating:** 4
**Confidence:** 3

**Summary:**

GLaVE-Cap is a Global-Local aligned video captioning framework that generates fine-grained and contextually consistent descriptions by bridging local and global information. To support evaluation and training, the authors construct GLaVE-Bench, and release a large-scale dataset GLaVE-1.2M containing 16K detailed captions and 1.2M QA pairs. Experiments on multiple state-of-the-art video-language models, including GPT-4o and Qwen2.5-VL-72B, demonstrate that GLaVE-Cap achieves superior fine-grained understanding and strong generalization across diverse benchmarks.

**Strengths:**

1. The paper proposes GLaVE-Cap, which combines vision-expert tracking with semantic-level global–local caption alignment, representing a creative bridge between traditional CV modules and large vision-language models.
2. It contributes two large-scale datasets, GLaVE-1.2M and GLaVE-Bench, enabling systematic and fine-grained evaluation across multiple video captioning dimensions, with extensive experiments on strong VLM backbones showing consistent gains.
3. The paper is well written and supported by clear figures and ablations, making a complex multi-stage pipeline understandable and highlighting the empirical value of detailed video captioning.

**Weaknesses:**

1. The GLaVE-Cap pipeline is overly complex and relies on multi-stage, caption-level integration rather than end-to-end feature fusion. This design increases the system’s dependency on intermediate textual outputs, which may propagate or amplify errors from earlier stages, and limits the model’s ability to maintain visual consistency across frames. As a result, the framework sacrifices efficiency and holistic visual-text alignment.

**Questions:**

1. Case studies mainly involve examples with only one salient object in motion. How would the proposed method behave when multiple objects move simultaneously within the same scene, or when multiple events occur concurrently in overlapping spatial regions? Would the model remain stable and consistent under such conditions?

2. Why did the authors choose to separately integrate Grounding DINO and SAM 2 instead of directly using Grounded-SAM 2, which already provides joint grounding and segmentation capabilities?

3. The method assumes that the VLM can reliably align each overview sentence with a specific keyframe range and then uses this alignment to drive downstream scene segmentation. However, the model only sees sampled keyframes and may propagate hallucinated boundaries.

---

> ### Author Response · Authors · 2025-11-18
> **Response to Reviewer PK8P Part 1**
>
> **[Weakness 1 Part 1] The GLaVE-Cap pipeline is overly complex and relies on multi-stage, caption-level integration rather than end-to-end feature fusion. This design increases the system's dependency on intermediate textual outputs, which may propagate or amplify errors from earlier stages, and limits the model's ability to maintain visual consistency across frames. As a result, the framework sacrifices efficiency and holistic visual-text alignment.**
>
> - The discussion of computational efficiency
>
> Our framework is deliberately designed to incur moderate computational overhead to achieve significantly higher video utilization efficiency and descriptive granularity. We will next provide detailed experimental analysis and metrics regarding computational cost.
>
> In practice, the time overhead of applying masks with Grounding DINO and SAM2 accounts for less than 1% of the time required for caption generation by the VLMs, and is therefore considered negligible. Regarding VLM inference cost, many factors influence actual runtime, including GPU architecture, operating system, and potential API calls to access the models. Therefore, to provide a more stable and comparable measurement of computational cost, we use "Consume Token", defined as Input Tokens plus 4 × Output Tokens, following standard API pricing conventions. To measure information richness, we employ GPT-4o to count the number of unique QA pairs that each caption can support across four categories (Scene, Object, Person, and Other), with their sum referred to as "Total Info." The resulting metric, "Info Cost" (Consume Token / Total Info), reflects the average token consumption required to generate each VQA instance. We conduct experiments on all 55 videos in the GLaVE-Cap dataset using Qwen2.5VL-72B as the VLM backbone. The table below reports the average Consume Token, Total Info per video, and the resulting Info Cost, which represents the average token cost per VQA instance.
>
> | Method | Consume Token | #Scene | #Object | #Person | #Other | Total Info | InfoCost (Token/QA) |
> |:----|:---:|:---:|:---:|:---:|:---:|:---:|:---:|
> | LVD-2M | 17,842 | 10.49 | 10.38 | 12.62 | 17.23 | 50.72 | 351 |
> | AuroraCap | 81,754 | 5.04 | 4.27 | 6.16 | 8.05 | 23.52 | 3,475 |
> | Vript | 34,632 | 14.29 | 12.47 | 14.84 | 17.89 | 59.49 | 582 |
> | LLavaVideo | 89,779 | 12.09 | 12.44 | 16.25 | 19.65 | 60.43 | 1,485 |
> | ShareGPT4Video | 72,823 | 11.55 | 12.45 | 15.84 | 19.20 | 59.04 | 1,233 |
> | Video Input | 124,049 | 5.80 | 7.05 | 7.22 | 10.49 | 30.56 | 4,059 |
> | GLaVE-Cap | 346,353 | 44.91 | 45.42 | 35.51 | 37.15 | 162.99 | 2,124 |
>
> Our framework strategically trades off computational cost against data efficiency and annotation quality: it incurs moderate computational overhead to achieve significantly higher video utilization efficiency and descriptive granularity. Specifically, GLaVE-Cap exhibits a lower InfoCost than baseline methods such as AuroraCap and Video Input (which directly captions videos using a VLM), while incurring approximately 50% higher InfoCost compared to existing state-of-the-art video captioning methods like ShareGPT4Video and LLaVA-Video. However, it generates 150% more unique video question-answer pairs per video, reflecting substantially improved video utilization efficiency. This makes our approach particularly advantageous in data-scarce scenarios, where access to large volumes of raw video is limited. By extracting far more information from each video, our method reduces reliance on large-scale video datasets. Furthermore, our approach achieves significantly superior descriptive granularity compared to other detailed captioning techniques. This capability enables the generation of video-QA pairs that support more fine-grained vision-language alignment, which in turn contributes to enhanced downstream performance, as validated in our main experiments.

---

> ### Author Response · Authors · 2025-11-18
> **Response to Reviewer PK8P Part 2**
>
> **[Weakness 1 Part 2] The GLaVE-Cap pipeline is overly complex and relies on multi-stage, caption-level integration rather than end-to-end feature fusion. This design increases the system's dependency on intermediate textual outputs, which may propagate or amplify errors from earlier stages, and limits the model's ability to maintain visual consistency across frames. As a result, the framework sacrifices efficiency and holistic visual-text alignment.**
>
> - The discussion of hallucination propagation issue
>
> Our intermediate local-caption generation stage produces descriptions for short video clips, and thus cannot be directly evaluated using standard hallucination benchmarks such as VideoHallucer, which are designed for whole-video captioning. Nonetheless, we can reasonably infer hallucination rates at each stage of our multi-stage pipeline. Specifically, our pipeline consists of three stages: (1) vision-expert–based mask annotation, (2) local caption generation that processes masked keyframes and summarizes their visual content, and (3) scene segmentation followed by global summarization, which are purely text-based. Through manual inspection, we found that vision experts (Grounding DINO + SAM2) produce highly accurate object masks with virtually no boundary errors, making this stage an unlikely source of hallucination. As shown in Appendix G, our final outputs on the VideoHallucer subset do not exhibit higher hallucination rates than competing methods, and the HHEM leaderboard further confirms that SOTA LLMs used for processing text introduce negligible hallucinations. We therefore conclude that hallucinations primarily arise in the detailed generation phase of local captioning, rather than being introduced by the other two stages, and the overall hallucination level also does not accumulate despite the presence of multiple stages.

---

> ### Author Response · Authors · 2025-11-18
> **Response to Reviewer PK8P Part 3**
>
> **[Question 1] Case studies mainly involve examples with only one salient object in motion. How would the proposed method behave when multiple objects move simultaneously within the same scene, or when multiple events occur concurrently in overlapping spatial regions? Would the model remain stable and consistent under such conditions?**
>
> We further showcase two additional challenging sports-broadcast videos with dense camera motion and overlapping actions in GLaVE-Bench (Appendix N, p.26, Fig. 11). In the sprint example, the model reliably tracks rapid running athletes and provides precise descriptions for each athlete. In the soccer example, GLaVE-Cap remains stable and consistently captures the main events, such as passes and offensive intent, even if there are over ten people doing different actions.
>
> **[Question 2] Why did the authors choose to separately integrate Grounding DINO and SAM 2 instead of directly using Grounded-SAM 2, which already provides joint grounding and segmentation capabilities?**
>
> To clarify, our implementation does leverage the integrated Grounded-SAM 2 framework, and we did not separately integrate the two models. Our contribution lies in the adaptation of its official tracking_demo_with_continuous_id code to our keyframe-based pipeline. The core modification was shifting its input from processing the raw video at a fixed step to operating on our pre-extracted keyframes. This adaptation reduces computational cost and is crucial for generating stable object masks in our setting.
>
> To prevent any misunderstanding, we will update the manuscript (Line 243) to precisely state: "We adapt the method provided by Grounded-SAM 2, which integrates Grounding DINO and SAM 2's image predictor to extract bounding boxes and object masks for each keyframe." The subsequent paragraphs (Lines 244-250) then detail how we utilize this adapted tracking mechanism to effectively manage newly appearing objects across keyframes, ensuring consistent and accurate tracking throughout the video.
>
> **[Question 3] The method assumes that the VLM can reliably align each overview sentence with a specific keyframe range and then uses this alignment to drive downstream scene segmentation. However, the model only sees sampled keyframes and may propagate hallucinated boundaries.**
>
> Our pipeline is designed with a clear separation between global context injection and scene segmentation to mitigate such risks. The process is as follows:
> - Overview Captioning: The overview caption is generated by providing the VLM with all video keyframes at once. This serves as a high-level semantic guide, offering global context.
> - Local Captioning: Each local caption is then generated based on the keyframes from its corresponding video clip. During this step, the overview caption is provided only as auxiliary context to help maintain narrative consistency across clips, but they do not dictate the semantic content of the local descriptions.
> - Scene Segmentation: Crucially, the final scene segmentation is not directly derived from the overview caption or its potentially noisy keyframe alignments. Instead, it is obtained through a standalone semantic analysis of the complete sequence of local captions, which inherently reflect the natural progression of scenes and events.
> In other words, our method establishes the keyframe-grounded local captions as the cornerstone; the overview caption plays an auxiliary role in their generation, and the final scene segmentation is derived directly from this reliable sequence of local captions. This architecture ensures that any inaccuracies in the overview's temporal boundaries are confined to their role as contextual hints and do not directly affect the semantic content of the local captions or the final segmentation boundaries. The reliance on the full, grounded sequence of local captions for segmentation provides a robust mechanism that prevents error propagation and ensures stable results.

---

> > ### Comment · Reviewer_PK8P · 2025-11-22
> >
> > Thank you for the detailed and well-organized rebuttal. The clarifications are appreciated, and the empirical contribution is clear. However, I still believe the main contribution is primarily engineering-oriented rather than methodological, so I will keep my current score.

---

> > > ### Author Response · Authors · 2025-11-23
> > > **On the Rebuttal of "The Main Contribution is Primarily Engineering-Oriented Rather Than Methodological"**
> > >
> > > Our main contribution is not merely engineering-oriented but methodological. **It lies in a thorough analysis of the fundamental limitations of video detailed captioning, which led to the design of a novel and principled architecture that establishes a new baseline for this task.** We will elaborate on these aspects in detail below.
> > > - **New perspective on video detailed captioning:** Previous works primarily focus on the summarization stage and rely on prompt design to ensure the quality of local captions, ignoring the potential of utilizing explicit strategies to further improve the quality. However, we argue that local captions, as the foundation of the summarization process, play a critical role in shaping the overall quality of video captions. Our work is the first to explicitly enhance local caption quality as a central strategy for improving the level of detail in video descriptions.
> > > - **Introducing visual experts into video captioning:** Previous works simply use VLMs to generate local captions. However, VLMs often struggle to fully capture fine-grained information in complex videos, such as accurate object counting and spatial relationships. In our framework, we incorporate task-specialized vision expert models to address this limitation, not because they are universally better, but because they are particularly effective at addressing the specific challenges that VLMs fail to resolve. These experts provide structured visual priors that enhance the local captioning process, resulting in more accurate and detailed local captions, which in turn serve as more reliable inputs for the summarization stage.
> > > - **Decoupling dynamic and static information:** We believe that local captions, as the foundation of video captioning, should capture both rich dynamic-related information and detailed static scene content. However, prior works such as ShareGPT4Video only place emphasis on dynamic changes while neglecting static details. To address this issue, we propose a dual-stream structure to decouple dynamic and static visual information and capture them separately. This design enables our local captions to accurately reflect temporal variations while preserving fine-grained static details, ultimately resulting in more comprehensive and informative video descriptions.
> > > - **Semantic scene segmentation and summarization:** Rather than relying on low-level visual changes (e.g., PySceneDetect in Vript), we propose a semantic-based scene segmentation and summarization. This provides better scalability to longer videos, reduces redundancy, and produces more coherent global descriptions.
> > > - **Overview-guided consistency control:** We identify a core limitation of the local-to-global paradigm: isolated local captions often result in inconsistencies between their descriptions, which ultimately leads to a lack of coherence in the final video caption. Our overview injection mechanism reduces this by introducing top-down semantic guidance, improving coherence with minimal overhead.

---

### Official Review · Reviewer_q9V7 · 2025-10-27

**Soundness:** 3
**Presentation:** 1
**Contribution:** 2
**Rating:** 4
**Confidence:** 5

**Summary:**

This paper introduces GLaVE-Cap, a novel framework for fine-grained video captioning that addresses limitations of existing local-to-global captioning paradigms. The method integrates two core modules: TrackFusion, which uses vision experts and a dual-stream architecture to generate detailed and consistent local captions, and CaptionBridge, which aligns local and global captioning through context injection and adaptive scene-level summarization. The authors also contribute GLaVE-Bench, a comprehensive evaluation benchmark with significantly more queries per video than existing datasets, and GLaVE-1.2M, a large-scale training dataset with fine-grained captions and QA pairs. Extensive experiments demonstrate state-of-the-art performance across multiple benchmarks, and ablation studies validate the effectiveness of the proposed components.

**Strengths:**

1. GLaVE-Bench and GLaVE-1.2M are valuable contributions, offering multi-scene videos with dense annotations that support more reliable and fine-grained evaluation and training.
2. The method achieves SOTA results across multiple benchmarks and generalizes well across different VLM backbones, demonstrating high performance.

**Weaknesses:**

1. The core "local-to-global" paradigm itself is not new, and the individual components (using vision experts for grounding, dual-stream processing, global context injection) have been explored in related forms in prior image and video understanding works. The primary novelty lies in the specific integration and orchestration of these ideas into a cohesive pipeline for this task.
2. The reliance on multiple vision experts and a multi-stage pipeline (e.g., for object tracking, dual-stream captioning, and scene segmentation) is computationally expensive, which may limit real-time or scalable deployment. Furthermore, the marginal performance improvement appears to come at the cost of a significant increase in computational load. While the paper focuses on captioning quality, it does not address critical deployment metrics like computational cost, latency, throughput, or efficiency. The authors should compare these metrics with those of other methods.
3. While hallucination is briefly evaluated, a more thorough analysis of how hallucinations propagate through the multi-stage pipeline is lacking.
4. The framework still relies on keyframe-based processing and may miss important visual elements in cluttered scenes, as acknowledged in the limitations. A shift to object-centric captioning is suggested but not implemented.

**Questions:**

1. The dual-stream structure in TrackFusion is a key component. Did you experiment with merging the dynamic and static streams in a different order or through a more integrated, iterative process rather than a final merge? What was the rationale for the chosen sequential approach?
2. The pipeline heavily relies on external vision experts (Grounding DINO, SAM 2). How critical is the accuracy of these models to the final caption quality? Have you analyzed how noise or errors in the object masks or tracking propagate through your system?
3. The multi-stage pipeline seems computationally intensive. Could you provide a rough estimate of the inference time and cost compared to a simpler, single-pass VLM approach for a typical video in your benchmark?

---

> ### Author Response · Authors · 2025-11-18
> **Response to Reviewer q9V7 Part 1**
>
> **[Weakness 1] The core "local-to-global" paradigm itself is not new, and the individual components (using vision experts for grounding, dual-stream processing, global context injection) have been explored in related forms in prior image and video understanding works. The primary novelty lies in the specific integration and orchestration of these ideas into a cohesive pipeline for this task.**
>
> First, we would like to clarify that our approach is not a "local-to-global" pipeline but a novel local–global interaction paradigm. The "Local-to-global" paradigm segments the video into clips, summarizes each clip, and then directly generates the final caption, overlooking global context injection and underutilizing local captions. To address these issues, our proposed local–global interaction paradigm uses an overview caption derived from the entire video to guide local caption generation, and then performs scene segmentation and summarization based on the semantic information of the local captions. In the main experiments, GLaVE-Cap significantly outperforms other video-detailed captioning methods, demonstrating the effectiveness of our local–global interaction paradigm.
>
> Then, our contribution lies in a thorough analysis of the limitations faced by the video detailed captioning task, based on which we carefully select and design the most suitable models and architectures. Rather than merely assembling known modules. We will elaborate on these aspects in detail below.
> - Introducing visual experts into video captioning: Previous works simply use VLMs to generate local captions. However, VLMs often struggle to fully capture fine-grained information in complex videos, such as accurate object counting and spatial relationships. In our framework, we incorporate task-specialized vision expert models to address this limitation, not because they are universally better, but because they are particularly effective at addressing the specific challenges that VLMs fail to resolve. These experts provide structured visual priors that enhance the local captioning process, resulting in more accurate and detailed local captions, which in turn serve as more reliable inputs for the summarization stage.
> - Decoupling dynamic and static information: We believe that local captions, as the foundation of video captioning, should capture both rich dynamic-related information and detailed static scene content. However, prior works such as ShareGPT4Video only place emphasis on dynamic changes while neglecting static details. To address this issue, we propose a dual-stream structure to decouple dynamic and static visual information and capture them separately. This design enables our local captions to accurately reflect temporal variations while preserving fine-grained static details, ultimately resulting in more comprehensive and informative video descriptions.
> - Semantic scene segmentation and summarization: Rather than relying on low-level visual changes (e.g., PySceneDetect in Vript), we propose a semantic-based scene segmentation and summarization. This provides better scalability to longer videos, reduces redundancy, and produces more coherent global descriptions.
> - Overview-guided consistency control: We identify a core limitation of the local-to-global paradigm: isolated local captions often result in inconsistencies between their descriptions, which ultimately leads to a lack of coherence in the final video caption. Our overview injection mechanism reduces this by introducing top-down semantic guidance, improving coherence with minimal overhead.

---

> ### Author Response · Authors · 2025-11-18
> **Response to Reviewer q9V7 Part 2**
>
> **[Weakness 2] The reliance on vision experts and a multi-stage pipeline is computationally expensive, which may limit real-time or scalable deployment. The marginal performance gains come at the expense of significantly increased computational overhead, and key deployment metrics such as latency, throughput, and efficiency are overlooked, with no comparisons provided to other methods.**
>
> Our framework is deliberately designed to incur moderate computational overhead to achieve significantly higher video utilization efficiency and descriptive granularity. Our method operates primarily at the data-processing stage to generate fine-grained caption and VQA annotations, which does not require real-time level latency. In downstream training and evaluation, models directly load the precomputed captions/VQA annotations without invoking GLaVE-Cap online; therefore, our method introduces no additional latency or throughput penalty. We will next provide detailed experimental analysis and metrics regarding computational cost.
>
> In practice, the time overhead of applying masks with Grounding DINO and SAM2 accounts for less than 1% of the time required for caption generation by the VLMs, and is therefore considered negligible. Regarding VLM inference cost, many factors influence actual runtime, including GPU architecture, operating system, and potential API calls to access the models. Therefore, to provide a more stable and comparable measurement of computational cost, we use "Consume Token", defined as Input Tokens plus 4 × Output Tokens, following standard API pricing conventions. To measure information richness, we employ GPT-4o to count the number of unique QA pairs that each caption can support across four categories (Scene, Object, Person, and Other), with their sum referred to as "Total Info." The resulting metric, "Info Cost" (Consume Token / Total Info), reflects the average token consumption required to generate each VQA instance. We conduct experiments on all 55 videos in the GLaVE-Cap dataset using Qwen2.5VL-72B as the VLM backbone. The table below reports the average Consume Token, Total Info per video, and the resulting Info Cost, which represents the average token cost per VQA instance.
>
> | Method | Consume Token | #Scene | #Object | #Person | #Other | Total Info | InfoCost (Token/QA) |
> |:----|:---:|:---:|:---:|:---:|:---:|:---:|:---:|
> | LVD-2M | 17,842 | 10.49 | 10.38 | 12.62 | 17.23 | 50.72 | 351 |
> | AuroraCap | 81,754 | 5.04 | 4.27 | 6.16 | 8.05 | 23.52 | 3,475 |
> | Vript | 34,632 | 14.29 | 12.47 | 14.84 | 17.89 | 59.49 | 582 |
> | LLavaVideo | 89,779 | 12.09 | 12.44 | 16.25 | 19.65 | 60.43 | 1,485 |
> | ShareGPT4Video | 72,823 | 11.55 | 12.45 | 15.84 | 19.20 | 59.04 | 1,233 |
> | Video Input | 124,049 | 5.80 | 7.05 | 7.22 | 10.49 | 30.56 | 4,059 |
> | GLaVE-Cap | 346,353 | 44.91 | 45.42 | 35.51 | 37.15 | 162.99 | 2,124 |
>
> Our framework strategically trades off computational cost against data efficiency and annotation quality: it incurs moderate computational overhead to achieve significantly higher video utilization efficiency and descriptive granularity. Specifically, GLaVE-Cap exhibits a lower InfoCost than baseline methods such as AuroraCap and Video Input (which directly captions videos using a VLM), while incurring approximately 50% higher InfoCost compared to existing state-of-the-art video captioning methods like ShareGPT4Video and LLaVA-Video. However, it generates 150% more unique video question-answer pairs per video, reflecting substantially improved video utilization efficiency. This makes our approach particularly advantageous in data-scarce scenarios, where access to large volumes of raw video is limited. By extracting far more information from each video, our method reduces reliance on large-scale video datasets. Furthermore, our approach achieves significantly superior descriptive granularity compared to other detailed captioning techniques. This capability enables the generation of video-QA pairs that support more fine-grained vision-language alignment, which in turn contributes to enhanced downstream performance, as validated in our main experiments.
>
> **[Question 3] The multi-stage pipeline seems computationally intensive. Could you provide a rough estimate of the inference time and cost compared to a simpler, single-pass VLM approach for a typical video in your benchmark?**
>
> Based on our experimental results in Weakness 2, we observe that, while a single-pass VLM approach consumes relatively few tokens per video, it generates limited-length outputs (approximately 1k–2k tokens per video). This constraint restricts the granularity and richness of the content, ultimately leading to a significantly higher InforCost (tokens per VQA) compared to our GLaVE-Cap approach. In other words, although GLaVE-Cap may require more tokens overall per video, it achieves a lower token cost per VQA by producing more informative and detailed captions, which enhances efficiency in terms of information density.

---

> ### Author Response · Authors · 2025-11-18
> **Response to Reviewer q9V7 Part 3**
>
> **[Weakness 3] A thorough analysis of how hallucinations propagate through the multi-stage pipeline is lacking.**
>
> Our intermediate local-caption generation stage produces descriptions for video clips, and thus cannot be directly evaluated using standard hallucination benchmarks such as VideoHallucer, which are designed for whole-video hallucination evaluation. Nonetheless, we can reasonably infer hallucination rates at each stage of our multi-stage pipeline. Specifically, our pipeline consists of three stages: (1) vision-expert–based mask annotation, (2) local caption generation that processes masked keyframes and summarizes their visual content, and (3) scene segmentation followed by global summarization, which are purely text-based. Through manual inspection, we found that vision experts (Grounding DINO + SAM2) produce highly accurate object masks with virtually no boundary errors, making this stage an unlikely source of hallucination. As shown in Appendix G, our final outputs on the VideoHallucer subset do not exhibit higher hallucination rates than competing methods, and the HHEM leaderboard further confirms that SOTA LLMs used for processing text introduce negligible hallucinations. We therefore conclude that hallucinations primarily arise in the detailed generation phase of local captioning, rather than being introduced by the other two stages, and the overall hallucination level also does not accumulate despite the presence of multiple stages.
>
> **[Weakness 4] The framework still relies on keyframe-based processing and may miss important visual elements in cluttered scenes. A shift to object-centric captioning is suggested but not implemented.**
>
> We leverage Grounding DINO and SAM2 to visually annotate keyframes. In both the local-caption and summary prompts, we explicitly instruct the model to separately describe each object and its actions using numeric IDs and masks, enabling more comprehensive coverage of important visual elements without requiring prior knowledge of target objects or questions.
>
> However, as noted in our limitations, brief or visually subtle elements may still be missed due to the keyframe sampling strategy, which might not capture every important moment in cluttered scenes. We plan to explore object-centric captioning in future work, which can provide more tailored descriptions based on known target objects and questions, potentially leading to more accurate reasonings and responses.
>
> **[Question 1] Did you experiment with merging the dynamic and static streams in a different order or through a more integrated, iterative process? What was the rationale for the chosen sequential approach?**
>
> We conducted experiments comparing different orders of merging the dynamic and static streams: (1) generating differential caption first and then utilizing the dynamic analysis to assist the static object description in static stream, and (2) the reverse order, generating detailed caption first and then using the static information to guide the dynamic analysis in dynamic stream. The experiment results are shown below:
>
> | Model | Acc ↑ | Hallucination ↓ | Not mention↓ |
> |:----|:---:|:---:|:---:|
> | Differential first | 62.39 | 19.39 | 22.60 |
> | Static first | 62.19 | 19.73 | 22.52 |
> | GLaVE-Cap-Qwen | 63.67 | 17.42 | 22.90 |
>
> The results show that both approaches achieve comparable performance, with slight variations. while our final GLaVE-Cap-Qwen approach outperforms both. This suggests that generating the two streams independently before merging yields the best results. In contrast, any strict sequential order, whether dynamic first or static first, tends to focus more on capturing either the static or dynamic aspects, potentially overlooking the other. This finally leads to less comprehensive descriptions and a slight drop in performance.
>
> **[Question 2] How critical is the accuracy of vision experts to the final caption quality? Have you analyzed how noise or errors in the object masks or tracking propagate through your system?**
>
> We rely on vision experts to provide numeric IDs and masks for major people and objects to explicitly track and differentiate them, enabling comprehensive and consistent descriptions. As concluded in SoM, the accuracy and completeness of these annotations directly impact the quality of local captions and ultimately the final captions.
>
> However, quantitatively evaluating noise or errors in object masks or tracking is challenging. We manually inspected the object masks and tracking results across the 55 videos in GLaVE-Bench and found that within continuous scenes, mask boundaries and tracking are almost error-free. However, during viewpoint changes such as close-ups or when a person temporarily leaves and re-enters the scene, vision experts may fail to maintain tracking and instead assign new labels, potentially introducing hallucinations of new characters into the final captions. We look forward to future advances in video segmentation to address these issues.

---

### Official Review · Reviewer_2Ct3 · 2025-10-28

**Soundness:** 2
**Presentation:** 2
**Contribution:** 2
**Rating:** 4
**Confidence:** 4

**Summary:**

The paper presents GLaVE-Cap, a global–local aligned video captioning framework that integrates vision experts through TrackFusion and summarizes local captions via CaptionBridge to improve contextual coherence and detail accuracy. It also introduces GLaVE-Bench and GLaVE-1.2M, two benchmarks for evaluating fine-grained captioning and question answering on multi-scene videos.

**Strengths:**

1. The paper identifies a meaningful gap in video captioning—the weak interaction between local and global descriptions—and introduces a novel global–local alignment framework to address it.
2. The proposed TrackFusion and CaptionBridge modules form a coherent architecture that links frame-level detail modeling with global summarization, while integrating “vision experts” for object-level reasoning.
3. The introduction of GLaVE-Bench and GLaVE-1.2M provides valuable large-scale benchmarks for fine-grained captioning and QA evaluation on multi-scene videos, which could benefit the broader research community.

**Weaknesses:**

1.	Although TrackFusion integrates object tracking and keyframe-based prompts, it lacks explicit mechanisms for multi-scale temporal reasoning. The model remains clip-level and fixed-window, limiting its ability to adapt to videos with diverse temporal dynamics.
2.	The proposed GLaVE-Bench dataset contains only 55 evaluation videos, which may be insufficient to support claims of generality and scalability.
3.	The evaluation omits comparisons with recent large-scale Video-Language Models (e.g., Qwen2-VL, InternVL3), making it unclear whether the proposed approach remains competitive in the current multimodal LLM landscape.

**Questions:**

See weakness.

---

> ### Author Response · Authors · 2025-11-18
> **Response to Reviewer 2Ct3**
>
> **[Weakness 1] Although TrackFusion integrates object tracking and keyframe-based prompts, it lacks explicit mechanisms for multi-scale temporal reasoning. The model remains clip-level and fixed-window, limiting its ability to adapt to videos with diverse temporal dynamics.**
>
> We achieve multi-scale temporal reasoning through overview caption injection. Specifically, we utilize keyframes from the entire video to generate an overview caption that captures the main events and content of the video. During local caption generation, this overview caption is provided as a prompt that offers the full video context to guide the process, enabling local-global multi-scale temporal reasoning. Thus, even though the visual input is at the clip level, our method can generate locally detailed captions that remain contextually coherent across the broader video timeline.
>
> Our method handles scenarios with diverse temporal dynamics through an adaptive keyframe selection mechanism. Instead of relying on a fixed window, we compute frame-level CLIP similarity to extract keyframes; when the video contains rapid temporal changes, it causes more significant visual changes, leading to a higher frequency of keyframe sampling. This allows us to adaptively select keyframes at different intervals within a video according to its semantic content and temporal changes instead of using a fixed window, which helps better capture diverse temporal dynamics. We will include this explanation in the revised manuscript Section 3.2 Line 235.
>
> **[Weakness 2] The proposed GLaVE-Bench dataset contains only 55 evaluation videos, which may be insufficient to support claims of generality and scalability.**
>
> Although we have only 55 videos, our total number of QA pairs reaches 6,491, which significantly surpasses previous commonly used benchmarks such as Video-MME (2,700 in total, with only 900 in the short subset) and MVBench (4,000). Within this limited set of videos, we carefully selected diverse topics to ensure data variety and benchmark generality. The videos in GLaVE-Bench cover all six major categories defined by Video-MME and include all but three of its 30 subcategories (basketball, magic show, and fashion), ensuring broad topical and fine-grained diversity.
> We acknowledge that the current size of GLaVE-Bench is relatively modest. In future work, we plan to further expand the dataset to cover more videos and scenarios, which will help strengthen the generality and scalability of our approach.
>
> **[Weakness 3] The evaluation omits comparisons with recent large-scale Video-Language Models (e.g., Qwen2-VL, InternVL3), making it unclear whether the proposed approach remains competitive in the current multimodal LLM landscape**
>
> The following table shows the evaluation results of recent large-scale Video-Language Models on GLaVE-Bench and Video-MME short subset:
>
> | | GLaVE-Cap | GLaVE-Cap | GLaVE-Cap | Video-MME |
> |:----|:---:|:---:|:---:|:---:|
> | Model | Acc ↑ | Hallucination ↓ | Not mention↓ | Acc ↑ |
> | Qwen2VL-72B | 33.63 | 24.98 | 55.17 | 49.89 |
> | InternVL3-78B | 46.69 | 21.81 | 40.29 | 58.11 |
> | GLaVE-Cap-Qwen | 63.67 | 17.42 | 22.90 | 73.19 |
>
> As mentioned in our introduction, recent large-scale Video-Language Models like Qwen2-VL and InternVL3 generate limited-length outputs in a single forward pass (approximately 1k–2k tokens), which constrains their ability to produce comprehensive and detailed descriptions. The substantial performance gap observed in our comparative experiments on GLaVE-Bench confirms this limitation, highlighting the advantage of our approach in generating more thorough video understanding.

---

> > ### Comment · Reviewer_2Ct3 · 2025-11-22
> >
> > Thank you for the detailed responses. I have just a few small follow-up questions.
> >
> > Q1: For the adaptive keyframe sampling, could you share a simple example or stats showing how the number of sampled keyframes differs for fast-changing vs. slow videos, or across different benchmarks?
> >
> > Q2: For the dataset, since each video has around 120 QA pairs, I’m curious how redundancy or overlapping questions are avoided.
> >
> > Finally, thanks for adding the comparison with Qwen2-VL and InternVL3, making a clear sense of GLaVE-Cap-Qwen's advantages.

---

> > > ### Author Response · Authors · 2025-11-23
> > > **Response to Reviewer 2Ct3's Follow-up Questions**
> > >
> > > **[Q1] For the adaptive keyframe sampling, could you share a simple example or stats showing how the number of sampled keyframes differs for fast-changing vs. slow videos, or across different benchmarks?**
> > >
> > > To demonstrate the efficacy of our adaptive keyframe sampling strategy, we present a comparative showcase on videos with varying temporal dynamics in Appendix O, p.27, Fig. 12.
> > >
> > > The upper portion features a sprint race (video 2yGaTOzaGIA in GLaVE-Bench). Here, even when the runner's core pose is stable, the drastically shifting background signals rapid motion, which triggers our CLIP sampler to adopt a dense sampling rate of one keyframe every 0.5 seconds to capture these essential dynamics. Conversely, the lower portion of Fig. 12 shows a segment from a slow-paced educational video (video HwnB8aCn8yE in GLaVE-Bench), featuring a mostly static content against a stable background. Here, the visual content remains largely consistent over time, allowing our sampler to select far fewer keyframes (about 4s per keyframe) while still preserving the semantic integrity of the narrative.
> > >
> > > **[Q2] For the dataset, since each video has around 120 QA pairs, I’m curious how redundancy or overlapping questions are avoided.**
> > >
> > > To effectively minimize redundancy across the approximately 120 QA pairs per video, we implemented a structured, multi-layered generation and verification process.
> > >
> > > Our approach begins at the scene level, where we first leverage GLaVE-Cap to generate mutually exclusive, fine-grained captions for each scene, establishing a foundation of non-overlapping semantic units. For each of these distinct scenes, we then prompt a powerful LLM to generate scene-level QA pairs under a strict constraint: for each of the 13 predefined attribute QA categories, at most one question-answer pair is created. This combination of scene independence and intra-scene category exclusivity fundamentally reduces the potential for redundancy.
> > >
> > > For global-level QA, which utilizes the complete video caption for 4 video-level predefined QA categories, like temporal understanding, we apply similarly strict quotas, limiting each question type to a maximum of three instances.
> > >
> > > Finally, every QA pair in GLaVE-Bench undergoes a rigorous manual review process. This crucial step is designed to identify and eliminate any residual redundancy, such as overlap between scene-level and global-level questions or semantic similarity between questions from different categories, ensuring the overall diversity and the QA uniqueness of the benchmark.

---

### Official Review · Reviewer_626Y · 2025-10-31

**Soundness:** 4
**Presentation:** 4
**Contribution:** 4
**Rating:** 4
**Confidence:** 4

**Summary:**

The paper proposes GLaVE-Cap, a novel global-local aligned video captioning framework integrating vision experts to improve fine-grained and contextually consistent video captions. It introduces two key modules:
- TrackFusion integrates Grounding DINO and SAM 2 for cross-frame object tracking and uses a dual-stream architecture (differential+detailed captions) to capture both static and dynamic details.
- CaptionBridge injects a global overview caption to guide local captioning and performs adaptive scene-level summarization for coherent video-level captions.

Additionally, the authors introduce:
- GLaVE-Bench, a new benchmark with ~5x more queries per video than existing datasets.
- GLaVE-1.2M, a large-scale dataset (16K videos, 1.2M QA pairs).

Experiments on GLaVE-Bench, Video-MME, VidCapBench, MVBench show state-of-the-art (SOTA) results. A student model (GLaVE-7B) fine-tuned on GLaVE-1.2M also outperforms baselines, validating dataset quality.

**Strengths:**

1. The paper has conceptual novelty by moving beyond the local-to-global paradigm by explicitly modeling bidirectional local-global interaction.

2. Integration of vision experts:
Clever combination of Grounding DINO+SAM2 for robust object tracking, plus dual-stream design effectively captures both dynamic and static details.

3. The paper proposes new benchmark and dataset.
GLaVE-Bench and GLaVE-1.2M fill an evaluation and data gap in fine-grained video captioning.
The QA-per-video (118 vs. 16) and multi-scene structure make it far more comprehensive.

4. Evaluations on 4+ major benchmarks with both closed (GPT-4o) and open (Qwen2.5-VL-72B) models show consistent SOTA performance.

5. Detailed ablations quantify the contribution of each module. Appendix provides thorough prompt designs and reproducibility materials.

6. Both quantitative and qualitative results (including user study in video generation) convincingly demonstrate improved caption granularity and contextual consistency.

**Weaknesses:**

1. **Limited novelty in components:**
Though integration is well-engineered, both TrackFusion (based on Set-of-Mark+vision experts) and CaptionBridge (context injection+summarization) build on known elements rather than introducing fundamentally new algorithms.

2. **Dependence on powerful LLMs:**
Most evaluations rely on GPT-4o and Qwen2.5-VL-72B. It's unclear how much gain comes from the model scale versus method design. Smaller models (e.g., Qwen2.5-VL-7B) perform notably worse.

3. Since GLaVE-Bench and GLaVE-1.2M are built using GLaVE-Cap captions, potential data leakage or alignment bias could inflate results, even if mitigated by re-captioning and human verification.

4. The framework requires multiple calls to large VLMs and expert models (Grounding DINO, SAM2), making it computationally heavy and less practical for large-scale or real-time applications.

5. While video generation results are interesting, the paper could better relate fine-grained caption quality to downstream generation metrics quantitatively.

**Questions:**

1. How is "adaptive scene segmentation" quantitatively validated beyond accuracy gains---are the segmentation boundaries human-verified?

2. Were hallucination rates or factual alignment metrics (e.g., VideoHallucer subset) computed per-stage to confirm reduced propagation?

3. Can the dual-stream design be generalized to text-only summarization tasks?

4. How much of the performance gain remains if using smaller backbones ($\leq$7B), without GPT-4o supervision?

5. How does GLaVE-Cap handle videos with dense camera motion or overlapping actions (e.g., sports broadcasts)?

6. Finally, I suggest resolving my concerns in Weaknesses.

---

> ### Author Response · Authors · 2025-11-18
> **Response to Reviewer 626Y Part 1**
>
> **[Weakness 1] Limited novelty in components: Though integration is well-engineered, both TrackFusion (based on Set-of-Mark+vision experts) and CaptionBridge (context injection+summarization) build on known elements rather than introducing fundamentally new algorithms.**
>
> Our key contribution lies in a thorough analysis of the limitations faced by the video detailed captioning task, based on which we carefully select and design the most suitable models and architectures. Parts of TrackFusion's structure are inspired by prior works, such as Set-of-Mark[1], and we found these works to be highly effective in assisting VLMs for local caption generation. However, our dual-stream structure, context injection mechanism, and adaptive scene segmentation are innovative structural improvements specifically designed to address the limitations of previous video detailed captioning methods. This principled framework demonstrates a deep understanding of the task and targeted technical innovations beyond simple integration. We will elaborate on these aspects in detail below.
> - Introducing visual experts into video captioning: Previous works simply use VLMs to generate local captions. However, VLMs often struggle to fully capture fine-grained information in complex videos, such as accurate object counting and spatial relationships. In our framework, we incorporate task-specialized vision expert models to address this limitation, not because they are universally better, but because they are particularly effective at addressing the specific challenges that VLMs fail to resolve. These experts provide structured visual priors that enhance the local captioning process, resulting in more accurate and detailed local captions, which in turn serve as more reliable inputs for the summarization stage.
> - Decoupling dynamic and static information: We believe that local captions, as the foundation of video captioning, should capture both rich dynamic-related information and detailed static scene content. However, prior works such as ShareGPT4Video only place emphasis on dynamic changes while neglecting static details. To address this issue, we propose a dual-stream structure to decouple dynamic and static visual information and capture them separately. This design enables our local captions to accurately reflect temporal variations while preserving fine-grained static details, ultimately resulting in more comprehensive and informative video descriptions.
> - Semantic scene segmentation and summarization: Rather than relying on low-level visual changes (e.g., PySceneDetect in Vript), we propose a semantic-based scene segmentation and summarization. This provides better scalability to longer videos, reduces redundancy, and produces more coherent global descriptions.
> - Overview-guided consistency control: We identify a core limitation of the local-to-global paradigm: isolated local captions often result in inconsistencies between their descriptions, which ultimately leads to a lack of coherence in the final video caption. Our overview injection mechanism reduces this by introducing top-down semantic guidance, improving coherence with minimal overhead.
>
> [1] Jianwei Yang, Hao Zhang, Feng Li, Xueyan Zou, Chunyuan Li, and Jianfeng Gao. Set-of-mark prompting unleashes extraordinary visual grounding in gpt-4v. arXiv preprint arXiv:2310.11441, 2023.
>
> **[Weakness 2 & Question 4] Dependence on powerful LLMs: Most evaluations rely on GPT-4o and Qwen2.5-VL-72B. How much of the performance gain remains if using smaller backbones (7B), without GPT-4o supervision?**
>
> We replaced the backbone with Qwen2.5VL-7B to evaluate performance using smaller backbones in GLaVE-Bench.
>
> | Model | Acc ↑ | Hallucination ↓ | Not mention ↓ |
> |:----|:---:|:---:|:---:|
> | AuroraCap | 31.09 | 21.14 | 60.58 |
> | LVD-2M | 42.57 | 25.30 | 43.01 |
> | LlavaVideo | 48.02 | 24.56 | 36.34 |
> | Vript | 51.43 | 23.33 | 32.92 |
> | ShareGPT4Video | 50.04 | 25.21 | 33.09 |
> | GLaVE-Cap | 49.85 | 25.68 | 32.92 |
> | GLaVE-Cap_auto_split | 55.20 | 23.49 | 27.85 |
> | GLaVE-Cap_gpt_split | 58.17 | 21.02 | 26.34 |
>
> When GLaVE-Cap relies solely on Qwen2.5VL-7B, it shows no clear performance advantage. Our analysis points to a key bottleneck: Qwen2.5VL-7B struggles to generate reliable scene segmentation from large volumes of local captions. This poor segmentation limits downstream effectiveness and suppresses gains. Note that our compared methods do not employ VLM-based scene segmentation, and thus are not affected by this specific bottleneck.
>
> Replacing model-generated splits with automatic PySceneDetect (as used in Vript) or GPT-4o-based segmentation significantly improves results. Even fully automatic splitting provided by PySceneDetect boosts performance noticeably, and a small amount of GPT-4o supervision (~5k tokens per video) further enhances it. This confirms that the segmentation quality is the main constraint under 7B.

---

> ### Author Response · Authors · 2025-11-18
> **Response to Reviewer 626Y Part 2**
>
> **[Weakness 3] Since GLaVE-Bench and GLaVE-1.2M are built using GLaVE-Cap captions, potential data leakage or alignment bias could inflate results, even if mitigated by re-captioning and human verification.**
>
> We confirm no data leakage, as re-captioning uses only raw videos and test videos are excluded during training the student model.
>
> Our benchmark for detailed video understanding necessitates fine-grained descriptions. GLaVE-Cap was selected as it provides +150% more detailed information compared to alternatives (Please refer to Common Response 2 for details), making it the only viable option to construct a comprehensive and precise evaluation. To address potential alignment bias as possible, we:
> - perform re-captioning and human verification to break the direct textual link to the original captions
> - design multi-perspective QAs per scene for each scene to ensure a comprehensive and non-bias assessment as possible
> - futher evaluate GLaVE-Cap on third-party benchmarks to demonstrate the performance gain is not from GLaVE-Cap's stylistic preferences
>
> **[Weakness 4] The framework requires multiple calls to large VLMs and expert models, making it computationally heavy and less practical for large-scale or real-time applications.**
>
> Our framework is deliberately designed to incur moderate computational overhead to achieve significantly higher video utilization efficiency and descriptive granularity. Our method operates primarily at the data-processing stage to generate fine-grained caption and VQA annotations, which does not requie real-time level latency. In downstream training and evaluation, models directly load the precomputed captions/VQA annotations without invoking GLaVE-Cap online; therefore, our method introduces no additional latency or throughput penalty. We will next provide detailed experimental analysis and metrics regarding computational cost.
>
> In practice, the time overhead of applying masks with Grounding DINO and SAM2 accounts for less than 1% of the time required for caption generation by the VLMs, and is therefore considered negligible. Regarding VLM inference cost, many factors influence actual runtime, including GPU architecture, operating system, and potential API calls to access the models. Therefore, to provide a more stable and comparable measurement of computational cost, we use "Consume Token", defined as Input Tokens plus 4 × Output Tokens, following standard API pricing conventions. To measure information richness, we employ GPT-4o to count the number of unique QA pairs that each caption can support across four categories (Scene, Object, Person, Other), with their sum referred to as "Total Info." The resulting metric, "Info Cost" (Consume Token / Total Info), reflects the average token consumption required to generate each VQA instance. We conduct experiments on all 55 videos in the GLaVE-Cap dataset using Qwen2.5VL-72B as the VLM backbone. The table below reports the average Consume Token, Total Info per video, and the resulting Info Cost, which represents the average token cost per VQA instance.
>
> | Method | Consume Token | #Scene | #Object | #Person | #Other | Total Info | InfoCost (Token/QA) |
> |:----|:---:|:---:|:---:|:---:|:---:|:---:|:---:|
> | LVD-2M | 17,842 | 10.49 | 10.38 | 12.62 | 17.23 | 50.72 | 351 |
> | AuroraCap | 81,754 | 5.04 | 4.27 | 6.16 | 8.05 | 23.52 | 3,475 |
> | Vript | 34,632 | 14.29 | 12.47 | 14.84 | 17.89 | 59.49 | 582 |
> | LLavaVideo | 89,779 | 12.09 | 12.44 | 16.25 | 19.65 | 60.43 | 1,485 |
> | ShareGPT4Video | 72,823 | 11.55 | 12.45 | 15.84 | 19.20 | 59.04 | 1,233 |
> | Video Input | 124,049 | 5.80 | 7.05 | 7.22 | 10.49 | 30.56 | 4,059 |
> | GLaVE-Cap | 346,353 | 44.91 | 45.42 | 35.51 | 37.15 | 162.99 | 2,124 |
>
> Our framework strategically trades off computational cost against data efficiency and annotation quality: it incurs moderate computational overhead to achieve significantly higher video utilization efficiency and descriptive granularity. Specifically, GLaVE-Cap exhibits a lower InfoCost than baseline methods such as AuroraCap and Video Input (which directly captions videos using a VLM), while incurring approximately 50% higher InfoCost compared to existing state-of-the-art video captioning methods like ShareGPT4Video and LLaVA-Video. However, it generates 150% more unique video question-answer pairs per video, reflecting substantially improved video utilization efficiency. This makes our approach particularly advantageous in data-scarce scenarios, where access to large volumes of raw video is limited. By extracting far more information from each video, our method reduces reliance on large-scale video datasets. Furthermore, our approach achieves significantly superior descriptive granularity compared to other detailed captioning techniques. This capability enables the generation of video-QA pairs that support more fine-grained vision-language alignment, which in turn contributes to enhanced downstream performance, as validated in our main experiments.

---

> ### Author Response · Authors · 2025-11-18
> **Response to Reviewer 626Y Part 3**
>
> **[Weakness 5] While video generation results are interesting, the paper could better relate fine-grained caption quality to downstream generation metrics quantitatively.**
>
> We evaluate the connection between fine-grained caption quality and downstream generation performance on the 15 sets of generated videos and their corresponding captions introduced in Appendix F. Specially, We first assess two types of similarity: (1) between the generated captions and the original video (denoted by the "-Caption" suffix), and (2) between the video generated from those captions and the original video (denoted by the "-Video" suffix). For each type of similarity, we then use the softmax of the similarity from GLaVE-Cap and ShareGPT4Video as the win rate. We use both VideoClip-XL and Qwen2.5-VL-72B to calculate the similarity to ensure robustness. The experiment results are shown below, and all metrics are averaged at the video-level.
>
> | Method | VIdeoCLIP-Similarity | VIdeoCLIP-Win rate | LVLM-Similarity | LVLM-Win rate |
> |:----|:---:|:---:|:---:|:---:|
> | ShareGPT4Video-Caption | 25.87 | 0.2940 | 69.33 | 0.2009 |
> | GLaVE-Cap-Caption | 26.95 | 0.7060 | 77.33 | 0.7991 |
> | ShareGPT4Video-Video | 82.28 | 0.2983 | 69.33 | 0.2013 |
> | GLaVE-Cap-Video | 85.30 | 0.7017 | 79.67 | 0.7987 |
>
> As shown in the results, both the fine-grained caption similarity and the downstream-generated video similarity consistently achieve win rates. This high level of agreement quantitatively demonstrates that our fine-grained captions not only describe the original video more accurately but also lead to higher-quality video generation. These results are further supported by and consistent with findings from our user study.
>
> **[Question 1] How is "adaptive scene segmentation" quantitatively validated beyond accuracy gains---are the segmentation boundaries human-verified?**
>
> We conducted manual inspections of the scene segmentation results for the 55 videos in GLaVE-Bench. (Note that our GLaVE-Bench dataset has been carefully manually refined to ensure that each scene is reasonable and valid. We specifically performed re-captioning on these 55 videos using Qwen2.5VL-72B as the backbone to detect segmentation errors). We categorized segmentation errors into three types: misaligned boundaries, missed splits, and redundant splits. These correspond to three issues: boundaries that are inaccurately segmented(Offset), splits that should have occurred but were missed(Missed), and redundant splits that should not have appeared(Redundant). The table below shows the occurrence of these three types of errors per video and the boundary error rate.
>
> | Method | Offset | Missed | Redundant | Boundary Error Rate |
> |:----|:---:|:---:|:---:|:---:|
> | Scene Segmentation | 0.09 | 0.44 | 0.73 | 18.2% |
>
> Although PySceneDetect has largely resolved boundary misalignment and virtually eliminated factual inaccuracies, the inherent subjectivity of segmentation still leads to results that are suboptimal for summarization:
> - Missed splits often occur when long logically segmentable continuous shots (such as a host walking through multiple rooms in 2QTiAmvygC4) remain undivided, potentially leading to over-condensation of information.
> - Redundant splits often arise in rapidly edited sequences (e.g., frequent close-ups in 9dQXiqS56Y4), where excessively brief segments (under 3 frames) disrupt narrative coherence.
>
> These findings confirm that even though semantic-based adaptive scene splitting strategies have shown significant improvement, a single fixed segmentation criterion still remains suboptimal due to the subjective nature of segmentation. Therefore, in future work, we will explore a tree-structured timeline that adaptively adjusts scene granularity based on keyframe span and content density, so as to better accommodate the subjective nature of scene segmentation.

---

> ### Author Response · Authors · 2025-11-18
> **Response to Reviewer 626Y Part 4**
>
> **[Question 2] Were hallucination rates or factual alignment metrics (e.g., VideoHallucer subset) computed per-stage to confirm reduced propagation?**
>
> Our intermediate local-caption generation stage produces descriptions for short video clips, and thus cannot be directly evaluated using standard hallucination benchmarks such as VideoHallucer, which are designed for whole-video captioning. Nonetheless, we can reasonably infer hallucination rates at each stage of our multi-stage pipeline. Specifically, our pipeline consists of three stages: (1) vision-expert–based mask annotation, (2) local caption generation that processes masked keyframes and summarizes their visual content, and (3) scene segmentation followed by global summarization, which are purely text-based. Through manual inspection, we found that vision experts (Grounding DINO + SAM2) produce highly accurate object masks with virtually no boundary errors, making this stage an unlikely source of hallucination. As shown in Appendix G, our final outputs on the VideoHallucer subset do not exhibit higher hallucination rates than competing methods, and the HHEM leaderboard further confirms that SOTA LLMs used for processing text introduce negligible hallucinations. We therefore conclude that hallucinations primarily arise in the detailed generation phase of local captioning, rather than being introduced by the other two stages, and the overall hallucination level also does not accumulate despite the presence of multiple stages.
>
> **[Question 3] Can dual-stream design be generalized to text-only summarization tasks?**
>
> Our dual-stream design relies on a natural separation in local captions: each contains two frames, allowing the model to distinguish static content in the current frame from changes across frames. Text-only summarization lacks such an inherent division between "state" and "transition", making it difficult to apply the same factorization. Thus, the dual-stream mechanism does not naturally extend to standard text-only summarization tasks.
>
> **[Question 5] How does GLaVE-Cap handle videos with dense camera motion or overlapping actions (e.g., sports broadcasts)?**
>
> Our method handles dense camera motion through an adaptive keyframe selection mechanism. We compute frame-level CLIP similarity to extract keyframes; when the video contains dense camera motion, it causes more significant visual changes, leading to a higher frequency of keyframe sampling. This allows us to adaptively select keyframes at different intervals within a video according to its semantic content and camera motion. We will include this explanation in the revised manuscript Section 3.2 Line 235.
>
> To ensure accurate coverage of complex scenes with overlapping actions, we employ a prompting mechanism orchestrated by numeric IDs and masks from vision experts. By explicitly directing the model in both local and summary prompts to describe each identified object individually and to note camera transitions, we enable a precise and disentangled narrative.
>
> We further showcase two additional challenging sports-broadcast videos with dense camera motion and overlapping actions in GLaVE-Bench (Appendix N, p.26, Fig. 11). In the sprint example, the model reliably tracks rapid panning and zooming, providing precise descriptions for each athlete. In the soccer example, GLaVE-Cap remains stable and consistently captures the main events, such as passes and offensive intent, even if there are over ten people doing different actions.

---

### Meta-Review · Area_Chair_kbuE · 2026-01-07

**Summary:**

The main worries were less about results and more about the story: several reviewers felt the method is mostly a strong engineering integration of known pieces. They also flagged the heavy multi-stage pipeline and reliance on big VLMs as costly and hard to scale. On the evaluation side, people questioned whether a 55-video benchmark is enough, whether building benchmarks/data using the method could bias results, and whether errors or hallucinations might propagate through stages (scene splits, keyframes, tracking) in ways not fully measured.

**Reviewer Concerns:**

The rebuttal addressed a lot concerns: they added comparisons to newer MLLMs (Qwen2-VL, InternVL3), gave cost/efficiency numbers (token-based “InfoCost”), validated scene segmentation with manual inspection stats, and connected caption quality to video generation with quantitative similarity wins. They also answered small-backbone questions by testing 7B and showing segmentation is the bottleneck, plus explained QA redundancy controls and stream-order ablations.

What’s still hanging: some reviewers still see the contribution as mainly engineering, stage-wise hallucination propagation is argued more than tightly quantified, 55 videos still feels small to some, and the method’s gains look less robust without strong models or high-quality splitting.

**Reviewer Scores:**

Reviewer 626Y likely increase score to 5 because most of their specific questions (segmentation validation, cost, 7B behavior, downstream link) got solid answers, even if novelty doubts remain.
Reviewer 2Ct3 likely goes 4 to 5 since missing comparisons were fixed and their follow-ups were answered concretely.
Reviewer q9V7 is a toss-up but I’d bet they stay 4 because they had high-confidence novelty and deployment concerns that aren’t fully erased.
Reviewer PK8P stays 4 because they explicitly said they’ll keep the score due to the “engineering-first” view.

---

### Decision · Program_Chairs · 2026-01-26

Reject